# Anytime-Valid Inference For Multinomial Count Data

**Michael Lindon**
Netflix
michael.s.lindon@gmail.com

**Alan Malek**
alan.malek@gmail.com

## Abstract

Many experiments compare count outcomes among treatment groups. Examples include the number of successful signups in conversion rate experiments or the number of errors produced by software versions in canary tests. Observations typically arrive in a sequence and practitioners wish to continuously monitor their experiments, sequentially testing hypotheses while maintaining Type I error probabilities under optional stopping and continuation. These goals are frequently complicated in practice by non-stationary time dynamics. We provide practical solutions through sequential tests of multinomial hypotheses, hypotheses about many inhomogeneous Bernoulli processes and hypotheses about many time-inhomogeneous Poisson counting processes. For estimation, we further provide confidence sequences for multinomial probability vectors, all contrasts among probabilities of inhomogeneous Bernoulli processes and all contrasts among intensities of time-inhomogeneous Poisson counting processes. Together, these provide an "anytime-valid" inference framework for a wide variety of experiments dealing with count outcomes, which we illustrate with several industry applications.

## 1  Introduction

Many experiments compare *count outcomes* among treatment groups (arms). For example, in online conversion experiments we seek the treatment group that achieves the greatest number of conversions (signups, purchases etc...). In software experiments, we seek the software version that produces the fewest number of errors [Lindon et al., 2022a, Kuo and Yang, 1996]. These experiments are frequently complicated in practice by time-variability over the course of the experiment caused by uncontrollable external factors. The Bernoulli success probability of conversion is often time-varying due to uncontrollable product changes and market conditions. The instantaneous rate of software errors is time-varying due to time-varying usage patterns of the software application among users. Fortunately, the time-variability is often common to all arms of the experiment. Therefore, the count outcomes among arms should be in some sense comparable under the null hypothesis that all arms behave equally.

In testing these hypotheses, it is desirable if a statistical test can both (i) detect large treatment effects early, and (ii) detect small treatment effects eventually. If there is a large negative effect then it is important to end the experiment as soon as possible to reduce risk to the experimental units and/or company. However, it is also important to detect small negative effects, as these can accumulate over time. Classical fixed-$n$ tests cannot satisfy both of these objectives, as there is only one opportunity to perform the test. Performing the test earlier achieves objective (i), but doesn't achieve objective (ii) as it may be underpowered. Performing the test later achieves objective (ii), but doesn't achieve objective (i) and treatments may affect experimental units negatively for lengthy amounts of time. For this reason we provide sequential tests that allow experiments to be continuously monitored. We have included a discussion of fixed-$n$ testing vs sequential testing and corresponding literature review in the appendix.

36th Conference on Neural Information Processing Systems (NeurIPS 2022).

Sequential testing in experiments with Bernoulli outcomes is complicated in practice because failure outcomes are typically not directly observed. In many fixed-$n$ approaches, Bernoulli fails are often *inferred* from the absence of successes within an arbitrary window of time. This is not possible for real-time sequential testing when the raw data are streams of Bernoulli successes with associated timestamps. We are able to circumvent this issue by basing the sequential test solely on the counts of successes accumulated for each arm so far. This has the advantage of also removing the time-varying components in the success probabilities.

We begin by providing an anytime-valid inference approach for multinomial families based on a mixture martingale, which is widely applicable to experiments dealing with count outcomes. We build upon this test to provide sequential tests of time-inhomogeneous Bernoulli processes and time-inhomogeneous Poisson counting processes. The former is widely applicable to industry A/B experiments with Bernoulli outcomes, which we illustrate with a payments funnel experiment. The latter is widely applicable to industry A/B experiments which observe point processes in time, which we illustrate with an A/B test used to monitor the safe roll-out of a new software application version. Our results are general for an arbitrary number of arms.

## 2 A Sequential Multinomial Test

### 2.1 Theory

Our development of sequential tests for different kinds of count data begins with a sequential test for multinomial observations. The construction follows the following sequence of steps common in the literature [Shafer et al., 2011, Waudby-Smith and Ramdas, 2020, Howard et al., 2021]. Step 1: Define a relevant test statistic. Step 2: Show that the test statistic is a test martingale - a nonnegative supermartingale under the null hypothesis. Step 3: Use martingale inequalities to bound the frequentist Type I error probability uniformly over time below a desired level $u$. Step 4: Invert the sequential test based on the test martingale to obtain a confidence sequence with a time-uniform coverage guarantee of at least $1 - u$. Extensions to other kinds of count data, including Bernoulli, Binomial and Poisson counting processes, are then obtained by recognizing relationships that exist to the multinomial distribution.

Our test martingale is obtained through a mixture martingale construction, integrating the multinomial likelihood with respect to a Dirichlet mixture distribution [Kaufmann and Koolen, 2021]. The likelihood mixture is also the statistic used in mixture sequential probability ratio tests (mSPRT) [Wald, 1945]. Bayesians will immediately recognize this statistic as a Bayes factor [Jeffreys, 1935, Kass and Raftery, 1995], interpreting the Dirichlet mixture distribution as a prior distribution, which could be used for computing posterior probabilities. However, not wishing to confuse the reader, we emphasize that we are not developing a Bayesian procedure. We aim to develop a procedure for "anytime-valid" inference, providing $u$-level sequential tests and $1 - u$ confidence sequences. We will borrow some language from Bayesian analysis, but the frequentist Type I error and coverage guarantees *do not* depend on the particular choice of Dirichlet distribution. To avoid confusion, we encourage the reader to view the Bayes factor simply as a test statistic.

The test statistic can also be viewed as an *e-process* [Grünwald et al., 2021, Pérez-Ortiz et al., 2022, Hendriksen et al., 2021]. A review of these constructions and their relationships is discussed in Ramdas et al. [2022]. The purpose of this paper is not to study the theoretical properties of these constructions [Wald and Wolfowitz, 1948], but rather to develop anytime-valid tests for several challenging applications in modern online experimentation dealing with count outcomes.

For step 1, consider a sequence $\boldsymbol{x}_1, \boldsymbol{x}_2, \boldsymbol{x}_3, \ldots$ of independent $\mathrm{Multinomial}(1, \boldsymbol{\theta})$ random variables with $\boldsymbol{\theta} \in \triangle^d$, the $d - 1$ simplex. We use bold typeset to denote vectors. Under the null hypothesis, $M_0$, it is assumed that

$$\boldsymbol{x}_1, \boldsymbol{x}_2, \ldots | M_0 \stackrel{\text{i.i.d.}}{\sim} \mathrm{Multinomial}(1, \boldsymbol{\theta}_0). \tag{1}$$

To construct a model over alternatives, $M_1$, we consider the marginal distribution obtained from the following joint model

$$\boldsymbol{x}_1, \boldsymbol{x}_2, \ldots | \boldsymbol{\theta}, M_1 \stackrel{\text{i.i.d.}}{\sim} \mathrm{Multinomial}(1, \boldsymbol{\theta}), \tag{2}$$
$$\boldsymbol{\theta} | M_1 \sim \mathrm{Dirichlet}(\boldsymbol{\alpha}_0).$$

The following expressions reduce to simple forms when a uniform distribution over the simplex is used, achieved by setting $\alpha_{0,i} = 1$ for $i = 1, \ldots, d$, which may be a good default value. The Type I error and coverage guarantees are unaffected by this choice, but the expected stopping time of the sequential test is influenced. The optimal choice of this parameter depends upon the value of $\boldsymbol{\theta}$ under the alternative, which is unknown. Stopping times under large departures from the null are reduced by using the uniform distribution. Whereas stopping times under very small departures from the null are reduced by concentrating the Dirichlet distribution about $\boldsymbol{\theta}_0$, which can be achieved with the choice $\alpha_{0,i} = k\theta_{0,i}$ for a concentration parameter $k \in \mathbb{R}^+$. In general, the average stopping time is reduced by matching the Dirichlet distribution to the distribution of effects upon which these sequential tests will be performed.

Let $S_i^n = \sum_{j=1}^n x_{j,i}$ and $\boldsymbol{S}_n = (S_1^n, \ldots, S_d^n) \in \mathbb{R}^d$. In addition, let $|\boldsymbol{v}| = \sum_i v_i$ denote the element-wise sum of a vector $\boldsymbol{v}$, $\boldsymbol{v^w} = \prod_i v_i^{w_i}$ to denote element-wise exponentiation of two vectors $\boldsymbol{v}$ and $\boldsymbol{w}$, and $\mathrm{Beta}$ to denote the multivariate Beta function $\mathrm{Beta}(\boldsymbol{v}) := (\prod_i \Gamma(v_i))/\Gamma(\sum_i v_i)$. The resulting Bayes factor comparing models $M_1$ to $M_0$ is given by

$$BF_{10}(\boldsymbol{x}_{1:n}) = \frac{\mathrm{Beta}(\boldsymbol{\alpha}_0 + \boldsymbol{S}_n)}{\mathrm{Beta}(\boldsymbol{\alpha}_0)} \frac{1}{\boldsymbol{\theta}_0^{\boldsymbol{S}_n}}. \tag{3}$$

which appears as early as Good [1967] (derivation in Appendix A.4). In a Bayesian analysis, the Bayes factor multiplied by the prior odds gives the posterior odds of $M_1$ over $M_0$. In this work, we take the prior odds to be unity so that the terms Bayes factor and posterior odds can be used interchangeably.

In sequential applications, it often makes sense to compute Equation (3) recursively. Let $O_n(\boldsymbol{\theta}_0)$ denote the posterior odds at $n$, then

$$O_n(\boldsymbol{\theta}_0) = \frac{\mathrm{Beta}(\boldsymbol{\alpha}_{n-1} + \boldsymbol{x}_n)}{\mathrm{Beta}(\boldsymbol{\alpha}_{n-1})} \frac{1}{\boldsymbol{\theta}_0^{\boldsymbol{x}_n}} O_{n-1}(\boldsymbol{\theta}_0), \tag{4}$$

where $\boldsymbol{\alpha}_n = \boldsymbol{\alpha}_{n-1} + \boldsymbol{x}_n$ and $O_0(\boldsymbol{\theta}_0) = p(M_1)/p(M_0) = 1$. When $\alpha_{0,i}$ are integer valued, Equation (4) can be expressed in terms of the posterior mean $\mathbb{E}[\theta_i | \boldsymbol{x}_{1:n-1}]$ for each $\theta_i$ as

$$O_n(\boldsymbol{\theta}_0) = \prod_{i=1}^d \left( \frac{\mathbb{E}[\theta_i | \boldsymbol{x}_{1:n-1}]}{\theta_{0,i}} \right)^{x_{n,i}} O_{n-1}(\boldsymbol{\theta}_0). \tag{5}$$

Details are provided in Appendix A.5. Equation (5) yields an intuitive betting interpretation [Shafer, 2021]. It represents our bet against the null hypothesis $\boldsymbol{\theta}_0$ using a $\boldsymbol{\theta}$ learned from the data. In this case, our estimate of $\boldsymbol{\theta}$ based on the data is the posterior mean. If a uniform prior is used, our initial bet on each outcome $x_{1,i}$ is simply $1/d$, and our bet on each outcome $x_{n,i}$ is $(1 + S_i^{n-1})/(n + d)$. The dependence of $O_n(\boldsymbol{\theta}_0)$ on the observed data $\boldsymbol{x}_{1:n}$ is implicit in this notation, yet the null value $\boldsymbol{\theta}_0$ being tested is made explicit to aid the discussion of confidence sequences in Theorem 2.4.

Step 2 in our construction is to demonstrate that $O_n(\boldsymbol{\theta}_0)$ is a nonnegative supermartingale under the null hypothesis $M_0$.

**Theorem 2.1.** *Let $\boldsymbol{x}_1, \boldsymbol{x}_2, \ldots$ be a sequence of independent* $\mathrm{Multinomial}(1, \boldsymbol{\theta})$ *random variables with the filtration $\mathcal{F}_{n-1} = \sigma(\boldsymbol{x}_1, \boldsymbol{x}_2, \ldots, \boldsymbol{x}_{n-1})$ and consider the sequence of posterior odds $O_n(\boldsymbol{\theta}_0)$ defined in Equation (4) with $O_0(\boldsymbol{\theta}_0) = 1$. Then*

$$\mathbb{E}_{M_0}[O_n(\boldsymbol{\theta}_0) | \mathcal{F}_{t-1}] = O_{n-1}(\boldsymbol{\theta}_0). \tag{6}$$

The proof is found in Appendix A.6. Theorem 2.1 states that $O_n(\boldsymbol{\theta}_0)$ is a nonnegative martingale under the null hypothesis with respect to the canonical filtration.

Step 3 is to use the posterior odds to construct a test martingale.

**Theorem 2.2.** *Let $\boldsymbol{x}_1, \boldsymbol{x}_2, \ldots$ be a sequence of independent* $\mathrm{Multinomial}(1, \boldsymbol{\theta})$ *random variables and consider the sequence of posterior odds $O_n(\boldsymbol{\theta}_0)$ defined in Equation (4) with $O_0(\boldsymbol{\theta}_0) = 1$. Then*

$$\mathbb{P}_{\boldsymbol{\theta}=\boldsymbol{\theta}_0} (\exists n \in \mathbb{N} : O_n(\boldsymbol{\theta}_0) \geq 1/u) \leq u \tag{7}$$

*for all $u \in [0, 1]$.*

The proof is provided in Appendix A.7. The time-uniform bound presented in Theorem 2.2 controls the deviations of a stochastic process for all $t$ simultaneously and is essential for proving the correctness of sequential tests and verifying the optional stopping and optional continuation properties. It provides a valid stopping rule: reject the null at time $\tau = \inf\{n \in \mathbb{N} : O_n(\boldsymbol{\theta}_0) \geq 1/u\}$. Simply stated, a practitioner who rejects the null hypothesis as soon as the posterior odds become larger than $1/u$ faces a frequentist Type I error with probability of at most $u$. Shafer et al. [2011], Johari et al. [2021] bring this idea back to more familiar territory by constructing a *sequential p-value* by tracking the running supremum of the posterior odds and taking its inverse, or equivalently

$$p_0 = 1 \text{ and}$$
$$p_n = \min(p_{n-1}, 1/O_n(\boldsymbol{\theta}_0)).$$

It follows from this definition and equation (7) that

$$\mathbb{P}_{\boldsymbol{\theta}=\boldsymbol{\theta}_0}\left(\exists n \in \mathbb{N} : p_n \leq u\right) \leq u, \tag{8}$$

which is an easily digestible generalization of a fixed-$n$ p-value to sequential settings. Instead of holding only at some pre-specified $n \in \mathbb{N}$, this guarantee holds *for all* $n \in \mathbb{N}$. This construction is shown in Figure 1. A simulation empirically demonstrating the control of false positives under continuous monitoring relative to a $\chi^2$ test is shown in Appendix A.9.

Before completing Step 4, it is useful to show that this sequential test is not trivial. For this test to have utility it must possess the ability to control not only Type I errors, as in theorem 2.2, but also Type II errors. This is provided by the following theorem.

**Theorem 2.3.** *Let $\boldsymbol{x}_1, \boldsymbol{x}_2, \ldots$ be a sequence of independent* $\mathrm{Multinomial}(1, \boldsymbol{\theta})$ *random variables and consider the sequence of posterior odds $O_n(\boldsymbol{\theta}_0)$ defined in Equation (4) with $O_0(\boldsymbol{\theta}_0) = 1$. If $\boldsymbol{\theta} \neq \boldsymbol{\theta}_0$, then*

$$\frac{1}{n}\log O_n(\boldsymbol{\theta}_0) \to D_{KL}(\boldsymbol{\theta}||\boldsymbol{\theta}_0) \qquad a.s., \tag{9}$$

*where $D_{KL}(\boldsymbol{\theta}||\boldsymbol{\theta}_0)$ is the Kullback-Leibler divergence of the true multinomial distribution with true parameter $\boldsymbol{\theta}$ from the multinomial distribution under the null hypothesis with null parameter $\boldsymbol{\theta}_0$.*

The proof is given in Appendix A.8. Theorem 2.3 states that if the null hypothesis is not true, with $\boldsymbol{\theta} \neq \boldsymbol{\theta}_0$, then the Bayes factor will diverge to infinity and exceed the $1/u$ threshold in Theorem 2.2 (a.s.), or equivalently that the sequential $p$-value converges to zero and falls below the $u$ threshold (a.s.). In other words, this test is guaranteed to reject the null almost surely if the null is incorrect, which is considered to be *asymptotically power 1* by Robbins [1970]. This result follows simply from the posterior consistency of Bayes factors. We now state step 4 of the construction.

**Theorem 2.4.** *Let $\boldsymbol{x}_1, \boldsymbol{x}_2, \ldots$ be a sequence of independent* $\mathrm{Multinomial}(1, \boldsymbol{\theta})$ *random variables and consider the sequence of posterior odds for testing the null hypothesis $\boldsymbol{\theta} = \boldsymbol{\theta}_0$, $O_n(\boldsymbol{\theta}_0)$, defined in Equation (4) with $O_0(\boldsymbol{\theta}_0) = 1$. Let $C_n(u) = \{\boldsymbol{\theta} \in \triangle^d : O_n(\boldsymbol{\theta}) < 1/u\}$ denote the set of parameter vectors that would not be rejected by the test at the $u$ level, then*

$$\mathbb{P}_{\boldsymbol{\theta}}\left(\boldsymbol{\theta} \in C_n(u) \text{ for all } n \in \mathbb{N}\right) \geq 1 - u \tag{10}$$

*for all $u \in [0, 1]$.*

A simple corollary of theorem 2.4 is that $\mathbb{P}_{\boldsymbol{\theta}}\left(\boldsymbol{\theta} \in \bigcap_{n=1}^{\infty} C_n(u)\right) \geq 1 - u$. This result provides a confidence statement for sequentially estimating the true parameter vector $\boldsymbol{\theta}$ as the experiment progresses. The confidence set $C_n(u)$ for $\boldsymbol{\theta}$ is a convex subset of $\triangle^d$, with convexity following from the concavity of the multinomial log-likelihood. Confidence intervals on the individual elements of $\boldsymbol{\theta}$ can be obtained by projecting $C_n(u)$ onto the coordinate axes in the following manner.

**Corollary 2.5.** *For $C_n(u)$ as in Theorem 2.4, let*

$$j_{n,i}^+(u) = \sup\{\theta_i : \boldsymbol{\theta} \in C_n(u)\},$$
$$j_{n,i}^-(u) = \inf\{\theta_i : \boldsymbol{\theta} \in C_n(u)\},$$

*then*

$$\mathbb{P}_{\boldsymbol{\theta}}\left(\forall i : \theta_i \in \bigcap_{n=1}^{\infty} [j_{n,i}^-(u), j_{n,i}^+(u)]\right) \geq 1 - u. \tag{11}$$

$j_{n,i}^+(u)$ can be computed by solving the following convex optimization program

$$
\begin{aligned}
\max \quad & \theta_i \\
\text{s.t.} \quad & c + \log u \leq \sum_i S_i^n \log \theta_i \\
& \sum_i \theta_i = 1
\end{aligned}
\tag{12}
$$

where $c = \log \text{Beta}(\boldsymbol{\alpha}_0 + \boldsymbol{S}_n) - \log \text{Beta}(\boldsymbol{\alpha}_0)$. The constraints in the optimization program simply define $C_n(u)$. Similarly, $j_{n,i}^-(u)$ is obtained by minimizing $\theta_i$ over this set.

Lastly, the confidence set in Theorem 2.4 is equal to the Bayesian *support set* [Wagenmakers et al., 2022].

**Corollary 2.6.** *The Bayesian support set, defined as $\{\boldsymbol{\theta} \in \triangle^d : p(\boldsymbol{\theta}|\boldsymbol{x}_{1:n}, M_1) \geq up(\boldsymbol{\theta}|M_1)\}$ has time-uniform frequentist coverage of at least $1 - u$.*

The proof is a simple application of the Savage-Dickey density ratio $O_n(\boldsymbol{\theta}) = p(\boldsymbol{\theta}|M_1)/p(\boldsymbol{\theta}|\boldsymbol{x}_{1:n}, M_1)$ [Dickey, 1971].

## 2.2 Application: Sample Ratio Mismatch Tests

Comparing the counts of experimental units assigned to each arm of a multi-arm experiment can often detect bias and errors in the experiment. Most online controlled experiments follow simply randomized designs, whereby a new unit is randomly assigned to one of $d$ arms according to a vector of pre-specified probabilities $\boldsymbol{\theta}$. The assignment outcome for a new unit is independent of other units (individualistic), unit-level covariates, potential outcomes (unconfounded) and can therefore be summarized as an independent $\text{Multinomial}(1, \boldsymbol{\theta})$ random variable. Under these assumptions, the assignment mechanism can be considered ignorable when performing inference on causal estimands such as the average treatment effect [Imbens and Rubin, 2015].

Although simple in theory, the systems that perform assignments quickly grow in complexity as the number of concurrent experiments increases [Tang et al., 2010]. This increased complexity increases the risk of introducing bugs that cause departures from the intended assignment mechanism, breaking the assumption of ignorability and rendering causal estimates invalid. Zhao et al. [2016] provide an account of an incorrect hashing algorithm introducing bias into the assignment mechanism.

After assignment and measurement, data passes through processing and cleansing pipelines before analysis. If incorrect cleansing logic is applied, there is a risk that specific observations may be selectively removed, introducing a "missing not at random" missing data mechanism, rendering causal estimates invalid [Rubin, 1976]. Fabijan et al. [2019] describes an experiment in which units from the treatment arm were unintentionally removed, with the probability of their removal depending on their observed outcomes.

An arm with a surprisingly low or high number of units is usually symptomatic of an implementation error in the experiment and is colloquially referred to as a *sample ratio mismatch* (SRM) [Fabijan et al., 2019]. These errors can be caught by comparing the counts of experimental units in each arm against the intended assignment probabilities. It is now considered good practice to validate the experiment setup by performing a $\chi^2$ test, comparing the observed counts against the expected counts under the intended assignment mechanism [Chen et al., 2018]. However, the $\chi^2$ test is an example of a *fixed-n* test, providing statistical guarantees when performed *once*. Due to this limitation, it is typically performed after data collection and before analysis. To reveal a bug that renders an expensive experiment invalid only after the experiment is finished would be less than ideal. Ideally, SRMs are detected as early as possible so that the implementation error can be corrected before more units enter the experiment. This necessitates sequential testing. We provide a sequential multinomial test for testing a point null based on the counts of each outcome.

Figure 1 shows the application of the sequential multinomial test to a representative experiment exhibiting an SRM. In this example the intended assignment probabilities are $\boldsymbol{\theta}_0 = (0.1, 0.4, 0.5)$. These propensity scores are assumed to be known by the analyst and will subsequently be used to perform inference about average treatment effects. Due to a bug, either in randomization or data processing, some units are preferentially assigned to certain arms based on their covariates, or data is

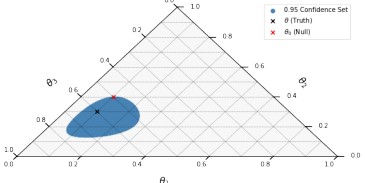

(a) Joint Confidence Set at $n = 144$

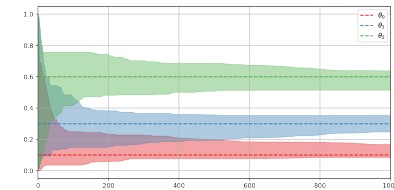

(b) Simultaneous Marginal Confidence Sequences

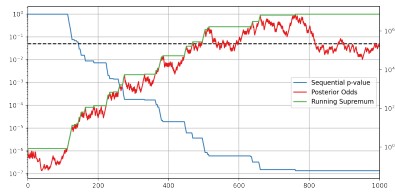

(c) Sequential $p$-value and Bayes Factor

Figure 1: (a) The confidence set $C_{144}(0.05)$ at $n = 144$ as defined in Theorem 2.4. The true $\boldsymbol{\theta} = (0.1, 0.3, 0.6)$ is marked with a black cross, and the null hypothesis $\boldsymbol{\theta}_0 = (0.1, 0.4, 0.5)$ is denoted with a red cross. (b) Simultaneous 0.95 confidence sequences that cover the individual elements of $\boldsymbol{\theta} = (0.1, 0.3, 0.6)$ obtained from Corollary 2.5 and computed via the optimization program in Equation (12). (c) (Left axis) Sequential $p$-value (blue) defined in Equation (8). Critical value $u = 0.05$ (dashed-black). $\boldsymbol{\theta} = (0.1, 0.3, 0.6)$, $\boldsymbol{\theta}_0 = (0.1, 0.4, 0.5)$. $p_n < 0.05$ for all $n \geq 144$. (Right axis) The posterior odds defined in Equation (3) (red), with the running supremum (green).

being lost not at random, resulting in observations arriving with probabilities $\boldsymbol{\theta} = (0.1, 0.3, 0.6)$. The sequential p-value is less than $0.05$ for all $n \geq 144$. This is the smallest $n$ for which $\boldsymbol{\theta}_0 \notin C_n(0.05)$.

## 3   Inhomogeneous Bernoulli Processes

Wald [1947] described an experiment to test the null hypothesis that two guns are equally precise. Two guns are fired at a target simultaneously for many attempts, and their success is recorded. Each attempt is modelled as a Bernoulli trial and, under the null hypothesis, the probability of each gun hitting the target on the same attempt is equal. However, the success probability varies across attempts due to gusty wind conditions. As the guns are shot simultaneously, the wind condition affects each gun equally. If the null is rejected, choosing the gun that obtained the greatest number of successes seems reasonable.

### 3.1   Theory

Suppose a new experimental unit is randomly assigned to one of $d$ experiment treatment groups (arms) at time $t$, according to assignment probabilities $\boldsymbol{\rho} \in \triangle^d$, and a Bernoulli outcome is observed. The Bernoulli probability for arm $i$ at time $t$ is parameterized by $p_i(t) = e^{\mu(t)} e^{\delta_i}$ so that the time-varying effect is multiplicative and common to all arms. The improvement of arm $j$ over arm $i$ at any time is then $p_j(t)/p_i(t) = \exp(\delta_j - \delta_i)$, and the difference on the log-scale is simply $\delta_j - \delta_i$. Suppose Bernoulli failures are ignored and arms are compared only through their counts of Bernoulli successes. The (conditional) probability that the next Bernoulli success comes from arm $i$ is

$$\theta_i = \frac{\rho_i e^{\delta_i}}{\sum_{j=1}^{d} \rho_j e^{\delta_j}}, \tag{13}$$

which is independent of the time-varying effect. The arm from which the next Bernoulli success arrives is, therefore, a $\mathrm{Multinomial}(1, \boldsymbol{\theta})$ random variable, and the counts of Bernoulli successes for

each arm are independent of the time-varying nuisance parameter $\mu(t)$. Framing the problem this way allows the sequential multinomial test to perform inference on $\boldsymbol{\delta}$. Simple hypotheses about $\boldsymbol{\delta}$ can therefore be translated into testing simple hypotheses about $\boldsymbol{\theta}$. Equality among success probabilities can be tested by simply testing the null multinomial hypothesis $\boldsymbol{\theta}_0 = \boldsymbol{\rho}$. The individual components $\delta_i$ are not identifiable, as adding a constant to each element results in the same $\boldsymbol{\theta}$, yet contrasts of the form $\sum_i a_i \delta_i$ for $\sum_i a_i = 0$ are identifiable.

Let $\sigma_{\boldsymbol{\rho}} : \mathbb{R}^d \to \triangle^d$ denote a generalization of the softmax function to include $\boldsymbol{\rho}$, with $\sigma_{\boldsymbol{\rho}}(\boldsymbol{\delta})_i$ equal to the right hand side of Equation (13).

**Theorem 3.1.** *Let $K_n(u) = \sigma_{\boldsymbol{\rho}}^{-1}(C_n(u))$, then*

$$\mathbb{P}[\boldsymbol{\delta} \in K_n(u) \text{ for all } n \in \mathbb{N}] \geq 1 - u \qquad (14)$$

The proof is a direct consequence of Theorem 2.4. The following corollary yields simultaneous confidence sequences for all contrasts.

**Corollary 3.2.** *Let $K_n(u) = \sigma_{\boldsymbol{\rho}}^{-1}(C_n(u))$ and $\mathcal{A}^d = \{\boldsymbol{a} \in \mathbb{R}^d : \sum_i a_i = 0\}$ denote the set of all $d$-dimensional contrasts. For all $\boldsymbol{a} \in \mathcal{A}^d$ define*

$$l_{n,\boldsymbol{a}}^+(u) = \sup\{\sum_i a_i \delta_i : \boldsymbol{\delta} \in K_n(u)\} \text{ and}$$

$$l_{n,\boldsymbol{a}}^-(u) = \inf\{\sum_i a_i \delta_i : \boldsymbol{\delta} \in K_n(u)\}.$$

*Then*

$$\mathbb{P}_{\boldsymbol{\theta}}\left(\forall \boldsymbol{a} \in \mathcal{A}^d : \sum_i a_i \delta_i \in \bigcap_{n=1}^\infty [l_{n,a}^-(u), l_{n,a}^+(u)]\right) \geq 1 - u.$$

The upper bound $l_{n,a}^+(u)$ is the solution to the convex optimization

$$
\begin{aligned}
\max \quad & \sum_i a_i \delta_i \\
\text{s.t.} \quad & c \leq \sum_i S_i^n \left( \delta_i + \log \rho_i - \log \sum_j \rho_j e^{\delta_j} \right),
\end{aligned} \qquad (15)
$$

where $c = \log \mathrm{Beta}(\boldsymbol{\alpha}_0 + \boldsymbol{S}_n) - \log \mathrm{Beta}(\boldsymbol{\alpha}_0) + \log u$. Convexity follows from the log-sum-exponential function. The lower bound $l_{n,a}^-(u)$ is the solution to the corresponding minimization problem. This is visualized in Figure 2.

A hypothesis can be rejected at the $u$ level as soon as the set that it defines fails to intersect with the confidence set $K_n(u)$. Note that $K_n(u_1) \subset K_n(u_2)$ for $u_1 > u_2$. To obtain a sequential $p$-value, we seek the largest $u$ such that the null is not rejected. That is, we seek the smallest set $K_n(u)$ such that there is a non-empty intersection with the subset of $\mathbb{R}^d$ defined by the hypothesis. This too can be achieved by a convex optimization program. One can simply maximize $u$ over the feasible set defined by the intersection of the $K_n(u)$ and the set defined by the hypothesis. Suppose one wishes to test the hypothesis $\delta_0 \geq \delta_1$ and $\delta_0 \geq \delta_2$. The sequential $p$-value at time $n$ is the inverse of the solution to the following convex program

$$
\begin{aligned}
\min \quad & q \\
\text{s.t.} \quad & c \leq \log(q) + \sum_i S_i^n \left( \delta_i + \log \rho_i - \log \sum_j \rho_j e^{\delta_j} \right) \\
& \delta_0 \geq \delta_1 \\
& \delta_0 \geq \delta_2
\end{aligned} \qquad (16)
$$

where $c = \log \mathrm{Beta}(\boldsymbol{\alpha}_0 + \boldsymbol{S}_n) - \log \mathrm{Beta}(\boldsymbol{\alpha}_0)$. A simulation study presenting sequential $p$-values and confidence sequences on $\boldsymbol{\delta}$ is given in Appendix A.11

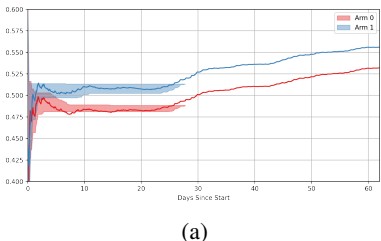 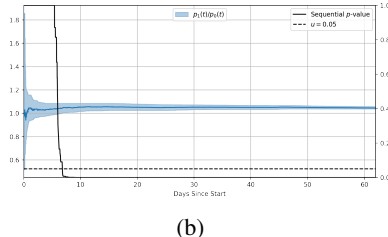

(a)                                                 (b)

Figure 2: (a) Assuming $p_1$ and $p_0$ are stationary and estimated using multinomial confidence sequences from from Corollary 2.5. (b) Assuming $p_1(t)$ and $p_0(t)$ are dynamic but with $p_1(t)/p_0(t) = \exp(\delta_1 - \delta_0)$ and estimating this quotient using the confidence sequences from equation Corollary 3.2. (Left axis) Confidence sequences are visualized with shaded regions and MLE estimates are visualized with solid lines. (Right axis) sequential $p$-value.

## 3.2 Application: Conversion Rate Optimization

Many modern experiments are designed to increase conversion rate and drive additional revenue. The term *conversion* is used to describe a Bernoulli trial such as user signup or purchase. The typical setup is that there are $d$ different *experiences* (such as different versions of a signup page), and each experience $i$ is assumed to have some conversion probability $p_i$. When a new user visits the experience, they either convert or they do not, and a successful or unsuccessful conversion event is logged. One might think that it would be easy to estimate $p_i$ sequentially using the multinomial confidence sequences of section 2 and the counts of successful and unsuccessful conversions of arm $i$. However, one soon runs into problems, as illustrated below.

The following case study comes from a signup funnel experiment at Netflix. It was hypothesized that adding additional methods of payments might increase the overall number of conversions. Alternatively, it could also be plausible that additional payment methods may simply cannibalize conversions from other payment methods, not increasing the overall number of conversions at all. An A/B test was created to test these hypotheses. The control group received the standard set of payment methods, while the treatment group were offered the standard set plus additional payment methods. To begin, let's first incorrectly assume that the success probabilities are constant. Figure 2 shows the application of the multinomial confidence sequences from Corollary 2.5 to separately estimate the conversion probabilities for arms 0 and 1. Note that the running intersection of anytime-valid confidence intervals becomes the empty set, and the MLE exits the confidence sequence. An empty intersection would be a rare event (with probability less than $\alpha$) if the assumption of constant probabilities in Section 2 were true. Instead, it indicates that the conversion probabilities are not constant but time-varying, invalidating a commonly made assumption in conversion rate experimentation.

Many online conversion experiments share similarities with Wald's gun-shooting example. The success probability of a conversion is often time-varying due to external factors such as the day of the week, recent product launches, or new promotions. These external factors likely affect all arms of the experiment equally, so it is reasonable to expect the time variation to be common to all. Moreover, it is often the case in these experiments that an unsuccessful signup (a 0 or "fail" Bernoulli outcome) is not directly observed. When a successful signup occurs, an event is typically logged, but if no signup occurs, then an unsuccessful signup must be inferred from the absence of a successful signup (typically through an ad-hoc definition such as "an experimental unit failed to convert within $T$ units of time since allocation"). This definition of conversion can make the analysis sensitive to the arbitrary choice of time window, and can present challenges to a sequential analysis if one must wait $T$ units of time before the first Bernoulli outcome is realized.

Instead, we propose using the assumptions of section 3, which only requires the counts of successful signups. The assumption of constant conversion probabilities is relaxed to a constant multiplicative difference between them, which is typically more realistic in practice. The confidence sequence on the multiplicative constant is shown in 2 (b). A winning arm can be declared in approximately one

week instead of nine, toward which all subsequent visitors can be directed, dramatically increasing signups relative to the fixed-$n$ experiment.

## 4 Inhomogeneous Poisson Counting Processes

### 4.1 Theory

A counting process is a stochastic process $\{N(t) : t \geq 0\}$ satisfying $N(0) = 0$, $N(t) \in \mathbb{N}_0$ and $N(s) \leq N(t)$ for $s \leq t$. The inhomogeneous Poisson counting process is defined by an *intensity function* $\lambda : \mathbb{R} \to \mathbb{R}^+$ that is locally integrable, $\int_B \lambda(t)dt \leq \infty$ for all bounded Borel measurable sets $B \in \mathbb{R}$, defining a measure $\Lambda(B) = \int_B \lambda(t)dt$ [Kingman, 1992]. For any collection of disjoint Borel measurable sets $B_1, B_2, \dots$ the inhomogeneous Poisson counting process has the property that $N(B_i)$ are independent $\mathrm{Poisson}(\Lambda(B_i))$ random variables. The inhomogeneous Poisson counting process can be defined in terms of an inhomogeneous Poisson point process on the real line by simply counting the number of points in a set. For our applications, these points correspond to times of events. At any time $t$ the probability density of the time-difference $s$ to the next point is given by $g(s) = \lambda(t + s) \exp(-\int_0^s \lambda(t + s)ds)$. The independent increments property implies a *memoryless* property of the process: the counts in the next time interval or waiting time until the next point are independent of the process history. This is a necessary property to establish the following theorem.

**Theorem 4.1.** *Consider $d$ inhomogeneous Poisson point processes indexed by $i \in \{1, \dots, d\}$ where process $i$ has intensity functions $\lambda_i(t) = \rho_i e^{\delta_i} \lambda(t)$. Let each point produced by process $i$ be marked with the corresponding process index $i$. At any time $t$, such as immediately after the previous point, the probability that the next point has mark $i$ is given by*

$$\theta_i = \frac{\rho_i e^{\delta_i}}{\sum_{j=1}^d \rho_j e^{\delta_j}}. \tag{17}$$

This gives the probability that the next point in time is from process $i$. The proof is given in the Appendix A.10. Theorem 4.1 states that the sequence of marks can be considered a sequence of $\mathrm{Multinomial}(1, \boldsymbol{\theta})$ random variables, allowing the sequential multinomial test to perform inference on $\boldsymbol{\delta}$. For example, a sequential test of equality among $d$ time-inhomogeneous Poisson point processes ($\lambda_i(t) = \lambda_j(t)$ for all pairs $i$ and $j$) can be obtained from the sequential multinomial test of the hypothesis $\boldsymbol{\theta}_0 = (1/d, \dots, 1/d)$. Once again, the total counts in each arm is a statistic that is independent of the time-varying nuisance parameter $\lambda(t)$. A simulation study presenting sequential $p$-values and confidence sequences on $\boldsymbol{\delta}$ is given in Appendix A.12

### 4.2 Application: Software Canary Testing

When continuously deploying new software to users, engineers often adopt the practice of *canary testing* [Lindon et al., 2022a, Schermann et al., 2018]. A canary test is a controlled experiment in which users are randomly assigned to the current software or a newer release candidate. The experimenter's goal is to study the performance of the release candidate in a production environment before releasing it globally, essentially acting as a quality control gate before full deployment. If the release candidate performs significantly worse, it is blocked, and developers must resolve the offending issues. This strategy helps to prevent bugs from reaching all users. However, performance regressions are still experienced by the users in the experiment. Reducing the risk for these users motivates the experiment's stoppage as soon as a performance regression is detected. Optional stopping requires performance to be measured in real-time and requires sequential testing methodology.

Performance regressions are measured in terms of the rates of *events*. These events could correspond to errors, failures, or any occurrence of interest sent to a central logging service by each instance of the software. The data naturally forms a marked 1-dimensional point process in time, recording the timestamp and type of event. The instantaneous rate is expected to be time-varying due to varying traffic and usage patterns.

The following example is taken from a canary test Netflix. This acts as a quality control gate in the software delivery process when releasing new versions of the client application. A regression that would have affected approximately 60% of all devices was detected with this methodology in less than one second. *Successful play starts* (SPS) are carefully monitored in this experiment between

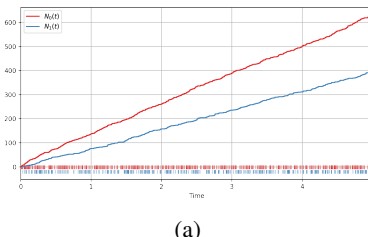
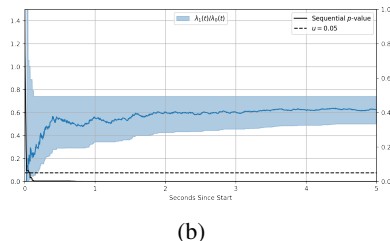

| (a) | (b) |

Figure 3: (a) Ruglplot shows the timestamps of SPS events being received, while solid lines show the counting processes for arms 0 and 1. (b) (Left axis) Shaded region shows the confidence sequence for the quotient of inhomogeneous Poisson process intensity functions $\lambda_1(t)/\lambda_0(t)$ obtained from Corollary 3.2. Blue solid line shows the MLE of $\lambda_1(t)/\lambda_0(t) = e^{\delta_1 - \delta_0}$. (Right axis) Black solid line shows sequential $p$-value.

the existing software and the release candidate. An SPS event is sent by the streaming application to the central logging system whenever a requested title successfully begins playback. If significantly fewer SPS events are being received from the treatment group running the release candidate, then it indicates there is an issue with the new software preventing some streams from starting. Figure 3 shows that the confidence sequence on $\lambda_1(t)/\lambda_0(t)$ falls below 1.0 in less than a second, indicating that the instantaneous rate of SPS events for arm 1 is less than the instantaneous rate of SPS events for arm 0. In this case, the canary test was aborted and the offending bug was identified, preventing a serious regression from being released to all users.

## 5   Conclusion

The contributions of this paper provide an "anytime-valid" approach to inference in several important applications dealing with count data. The anytime-valid guarantees permit experiments to be continuously monitored and enables optional stopping, which can greatly reduce both the time required for experiments to complete and the potential harm to units in the experiment. We first introduced a sequential test for Multinomial hypotheses using a mixture martingale construction, which has already proven to be an effective solution for rapidly detecting sample ratio mismatches in online controlled experiments. We then used this result to develop a sequential test of equality and contrasts in inhomogeneous Bernoulli processes, which has practically demonstrated dramatic speedups in decision-making in conversion experiments. Contrary to many widely used models, our approach does not assume that the conversion probabilities are constant, which is often violated in the real world. Moreover, only successful conversions, such as signups, are observed in practice. This is different from Bernoulli outcome models in which it is assumed that both successes and failures are observed. Our proposed sequential test enjoys the added convenience of only requiring successful Bernoulli outcomes to be observed. Lastly, we used the sequential Multinomial test to develop a sequential test for equality and contrasts in time-inhomogeneous Poisson counting processes. These play an important role in the monitoring of systems, such as data pipelines and software usage.

The confidence sequences provided in this paper allow inference to be made at any time and data-dependent stopping rules for ending an experiment. This in of itself can dramatically speed up the time to reach conclusions with valid statistical guarantees. An obvious extension to this work is to combine these confidence sequences with a strategy for adapting the assignment probabilities with the goal of assigning fewer units to suboptimal arms. The confidence sequences presented here can be used to construct an adaptive algorithm that identifies the best arm with tunably high probability through the least upper confidence bound (LUCB) algorithm, as has been done successfully in Howard and Ramdas [2022].

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
