# A  Appendix

## A.1  Limitations of Fixed-$n$ Testing

To quote Armitage [1993], "the classical theory of experimental design deals predominantly with experiments of predetermined size, presumably because the pioneers of the subject, particularly R. A. Fisher, worked in agricultural research, where the outcome of a field trial is not available until long after the experiment has been designed and started.". The author points out that many popular statistical tests are of the *fixed-n* or *fixed-horizon* kind, which operate just once on a complete dataset in a Neyman-Pearson type testing framework [Neyman et al., 1933]. The early development of these tests was driven by the wide variety of applications in which all observations arrive at the same time or when the experimenter is simply handed a complete dataset [Robbins, 1952]. Following this mode of one-time statistical analysis, these tests have been specifically optimized to maximize power at analysis time subject to a Type I error configuration. The classical solution to minimizing experiment cost is then to find a statistical test that provides the same Type I/II error guarantees at a smaller sample size, such as seeking the uniformly most powerful unbiased test within a particular class [Casella and Berger, 2002]. In many modern applications, however, data typically arrive in streams rather than in sets; observations often arrive in a sequence instead of simultaneously. Therefore, it makes sense that statistical tests optimized for a one-time analysis of a complete collection of observations may not be optimal in experiments where observations arrive sequentially.

There are practical difficulties in using fixed-$n$ tests in experiments where observations arrive sequentially. Consider these problems first from the perspective of hypothesis *testing*. The biggest drawback of using a fixed-$n$ test in a sequential application is that it can only be performed *once*. Fixed-$n$ tests are fine in applications where all observations arrive simultaneously or when the experimenter is handed a complete dataset, as there is only one possible opportunity to perform the test. However, if observations arrive in a sequence, the experimenter is presented with many opportunities to perform the test. Perform the test too early, and the Type II error probability will be high, resulting in many small effects being undetected. Perform the test too late, and the experiment may be more costly than is strictly necessary. Sample size calculations also fail to remedy these issues for the following reasons. Closed-form sample size expressions may not exist beyond trivial textbook models, the required inputs are frequently unknown, and most problematically, sample size calculations require the specification of a minimum detectable effect (MDE). The problem with specifying an MDE is that the experimenter, unwilling to sacrifice power even for small effect sizes, typically specifies these to be conservatively low, resulting in quadratic growth of the required sample size and a more costly experiment than strictly necessary (in particular, relative to a sequential design). For instance, consider one-sided "no-harm" testing applications where the goal is to detect (possibly small) adverse effects. Specifying a small MDE causes the required sample size to be large, resulting in large adverse effects remaining undetected for lengthy amounts of time and prolonged harm to the experimental units.

The latter example highlights a further practical difficulty with using fixed-$n$ tests in sequential applications from the *estimation* perspective. The experimenter is often curious about the current performance of each arm, stemming from the concern that an arm might have a substantial adverse effect on those assigned to it. To address this concern, the experimenter might wish to estimate the effect by computing a confidence interval. Unfortunately, the $1 - u$ confidence statement obtained by inverting an $u$-level fixed-$n$ test only holds at a fixed-$n$: it is only a one-time guarantee. The experimenter cannot hope to "monitor" the effect of each arm by computing multiple fixed-$n$ confidence intervals spaced out over different times, as their intersection does not have any coverage guarantee.

Instead of stopping the experiment at a predetermined sample size, it is more natural and useful for the stopping rule to be data-dependent in sequential applications. That is, to perform *optional stopping*. Stopping a test based on whether the data observed so far contains strong evidence for or against a hypothesis removes the need to perform a troublesome sample size calculation and allows the experiment to be terminated adaptively. If conclusions seem unclear at a chosen analysis time, such as confidence statements being insufficiently tight, then allowing the test to run for longer is also useful. That is, to perform *optional continuation*. Unfortunately, the Type I error and confidence guarantees from fixed-$n$ tests are not preserved under *optional stopping or continuation*.

Many experimenters do not specify a fixed sample size in advance simply because they have not made up their minds about the requirements of the experiment or the available resources [Anscombe, 1954]. This can lead to an invalid practice of "peeking" where a fixed-$n$ test is used to define a stopping rule, and estimands are monitored continuously by repeated fixed-$n$ confidence statements, a procedure that does not possess the statistical guarantees that an experimenter might naively expect [Johari et al., 2017]. Repeated applications of fixed-$n$ tests on accumulating sets of data result in ever-increasing Type I error probabilities [Armitage et al., 1969]. A stopping rule configured to stop sampling when a hypothesis is rejected by a fixed-$n$ test is guaranteed to reject the null, allowing experimenters to sample to a foregone conclusion [Anscombe, 1954, Kadane et al., 1996]. Consequently, the intersection of fixed-$n$ confidence sets is guaranteed to converge to the empty set.

## A.2 Solutions via Sequential Testing

Sequential designs remain the preferred form of scientific inquiry by many, so these experimenters would benefit greatly from the development of new statistical tests that support the desired operations of continuous monitoring with optional stopping and continuation. The solution presented here generalizes the frequentist guarantees already familiar to many by extending results to hold *for all n* instead of a *fixed-n*. We compute a *sequential p-value* such that the probability of this being less than $u$ *for any* $n \in \mathbb{N}$ is less than $u$. Similarly, we compute *confidence sequences*: a countable collection of sets such that the probability the estimand is covered *by all* sets, and hence their intersection, is greater than $1 - u$. We provide a review of the sequential testing literature in Appendix A.3.

There are numerous advantages to this approach. Confidence sets and $p$-values remain valid at all times, which enables experimenters to "check in" and *continuously monitor* the progress of their experiments. The ability to perform optional stopping allows developers to build a layer of automated stopping logic on top of experiments, reducing risk by quickly eliminating poorly-performing arms and terminating as soon as hypotheses have been rejected. This removes the need for human supervision and helps scale the number of experiments performed by automating their orchestration. This approach also appeals to both Bayesians and Frequentists: despite presenting the frequentist properties, it is fundamentally built upon a Bayes factor. Confidence sequences are constructed for the *vector* of parameters for all arms, providing simultaneous confidence sequences for all contrasts among arms in contrast to pairwise comparison tests, which require multiple testing corrections. Lastly, our approach is applicable to situations in which there is time-variability common to all arms, as we construct a test statistic that removes the time-varying components.

## A.3 Review of Sequential Testing

The earliest work is often attributed to Ville [1939] with the introduction of a *test martingale*. This object is a nonnegative supermartingale under the null hypothesis, and one can use martingale inequalities to construct sequential designs that control Type I error. Wald [1945] introduced the mixture sequential probability ratio test (mSPRT) for testing composite vs simple null hypotheses. The mSPRT can be viewed as a Bayes factor by interpreting the weight function used to integrate the likelihood ratio as a Bayesian prior over the alternative. Testing in a Bayesian framework via Bayes factors is attributed to Jeffreys [Jeffreys, 1935, Kass and Raftery, 1995]. Proofs of the validity of Bayes factors with nuisance parameters under varied interpretations of optional stopping are provided by Hendriksen et al. [2021]. The use of Bayes factors for sequential testing exists, therefore, both in a purely Bayesian framework from computing posterior probabilities over hypotheses and in Wald's mSPRT framework for obtaining frequentist error probabilities. Similarities and differences between both approaches are discussed at length in Berger et al. [1994, 1997, 1999]. Bayes factors have been used in the design of sequential clinical trials by Cornfield [1966]. Test martingales can be interpreted as Bayes factors and the inverse of the running supremum can be used to construct sequentially valid $p$-values [Shafer et al., 2011].

Johari et al. [2021] use Wald's mSPRT to construct *anytime-valid* inference for the difference in two Gaussian means with known variance by using the inverse of the running supremum of the mSPRT test martingale to construct a sequential $p$-value, and use the duality between $p$-values and confidence sets to construct confidence sequences for the difference. The "anytime-valid" namesake explicitly refers to the fact that this test is safe under optional stopping, in the sense that we may reject the null at the $u$-level as soon as the sequential $p$-value falls below $u$ without violating the Type I error guarantees. Similarly, confidence intervals have a $1 - u$ coverage guarantee at any time, allowing

the progress of a statistical test to be continuously monitored and making it robust to the human temptation to peek at results [Johari et al., 2017]. In contrast to our proposal for count data, however, this method requires Gaussian approximations based on central limit theorem arguments and requires a plugin estimators for unknown parameters. Although this method works well in practice, these approximations may not be justified at lower sample sizes, and so the sequential properties of this method may not be strictly guaranteed outside of Gaussian families. Confidence sequences appear as early as Darling and Robbins [1967]. Robbins [1970] showed that it is always possible to disprove a null hypothesis by sequentially collecting data until the null is rejected at the $u$-level by a fixed-$n$ frequentist test, regardless of the chosen value of $u$. This result follows as a consequence of the law of iterated logarithm. The anytime-valid approach through the use of sequential $p$-values and confidence sequences has been greatly extended by Howard et al. [2021], providing univariate nonparametric and nonasymptotic confidence sequences for broad classes of random variables. Confidence sequences for doubly robust causal estimands are presented in Waudby-Smith et al. [2021]. Confidence sequences for sampling without replacement are provided in Waudby-Smith and Ramdas [2020]. Anytime-valid $F$-tests and confidence sequences for subsets of linear regression coefficients are provided by Lindon et al. [2022b]. Design based confidence sequences for anytime-valid causal inference are provided by Ham et al. [2022].

Between fixed and anytime-valid/sequential testing are group sequential testing (GST) methods [Jennison and Turnbull, 1999]. In GST a finite and fixed number of analyses are planned as part of the design and are performed upon reaching the pre-specified sequence of sample sizes. There is the opportunity to reject the null hypothesis or continue to the next round at each analysis. GST only partially solves the optional stopping requirement and fails to solve the optional continuation requirement. Optional stopping is only partially solved because analyses can occur at pre-determined sample sizes, when practically the requirement is to perform analyses at pre-determined *times*. Suppose an experimenter wishes to perform analyses every day for a month. Due to varying traffic, there is no guarantee that the pre-determined sample sizes align with every day of the month. Optional continuation is not permitted as one loses the ability to collect more data beyond the final analysis. Silva and Kulldorff [2015] show that for every group sequential test, there exists a fully sequential test that is uniformly better in its ability to stop sooner.

### A.4 Derivation of Equation (3) (Bayes Factor)

The Bayes factor is defined as

$$BF_{01}(x_{1:n}) = \frac{p(\boldsymbol{x}_{1:n}|M_1)}{p(\boldsymbol{x}_{1:n}|M_0)} = \frac{\int p(\boldsymbol{x}_{1:n}|\boldsymbol{\theta}, M_1)p(\boldsymbol{\theta}|M_1)d\boldsymbol{\theta}}{p(\boldsymbol{x}_{1:n}|M_0)}. \tag{18}$$

Under the assumptions for $M_0$ and $M_1$ expressed in Equation (1) and Equation (2), we can show

$$
\begin{aligned}
p(\boldsymbol{x}_{1:n}|M_1) &= \int p(\boldsymbol{x}_{1:n}|\boldsymbol{\theta}, M_1)p(\boldsymbol{\theta}|M_1)d\boldsymbol{\theta} \\
&= \int \frac{\Gamma\left(\sum_{ij} x_{i,j} + 1\right)}{\prod_j \Gamma\left(\sum_i x_{i,j} + 1\right)} \prod_j \theta_j^{\sum_i x_{i,j}} \frac{\Gamma\left(\sum_j \alpha_{0,j}\right)}{\prod_j \Gamma(\alpha_{0,j})} \prod_j \theta_j^{\alpha_{0,j}-1} d\boldsymbol{\theta} \\
&= \frac{\Gamma\left(\sum_{ij} x_{i,j} + 1\right)}{\prod_j \Gamma\left(\sum_i x_{i,j} + 1\right)} \frac{\Gamma\left(\sum_j \alpha_{0,j}\right)}{\prod_j \Gamma(\alpha_{0,j})} \int \prod_j \theta_j^{\sum_i x_{i,j}+\alpha_{0,j}-1} d\boldsymbol{\theta} \\
&= \frac{\Gamma\left(\sum_{ij} x_{i,j} + 1\right)}{\prod_j \Gamma\left(\sum_i x_{i,j} + 1\right)} \frac{\Gamma\left(\sum_j \alpha_{0,j}\right)}{\prod_j \Gamma(\alpha_{0,j})} \frac{\prod_j \Gamma\left(\sum_i x_{i,j} + \alpha_{0,j}\right)}{\Gamma\left(\sum_{ij} x_{i,j} + \sum_j \alpha_{0,j}\right)} \quad \text{and} \\
p(\boldsymbol{x}_{1:n}|M_0) &= \frac{\Gamma\left(\sum_{ij} x_{i,j} + 1\right)}{\prod_j \Gamma\left(\sum_i x_{i,j} + 1\right)} \prod_j \theta_{0,j}^{\sum_i x_{i,j}}.
\end{aligned}
\tag{19}
$$

The result follows from cancelling terms in numerator and denominator:

$$BF_{01}(x_{1:n}) = \frac{p(\boldsymbol{x}_{1:n}|M_1)}{p(\boldsymbol{x}_{1:n}|M_0)} = \frac{\Gamma(\sum_{j=1}^d \alpha_{0,j})}{\Gamma(\sum_{j=1}^d \alpha_{0,j} + \sum_{i=1}^n x_{i,j})} \frac{\prod_{j=1}^d \Gamma(\alpha_{0,j} + \sum_{i=1}^n x_{i,j})}{\prod_{j=1}^d \Gamma(\alpha_{0,j})} \frac{1}{\prod_{j=1}^d \theta_{0,j}^{\sum_{i=1}^n x_{i,j}}}.$$

(20)

## A.5 Derivation of Equation (4) (Sequential Posterior Odds Updating)

The posterior odds in favor of $M_1$ to $M_0$ after observing $\boldsymbol{x}_{1:n}$ is defined as

$$\begin{aligned}
\frac{p(M_1|\boldsymbol{x}_{1:n})}{p(M_0|\boldsymbol{x}_{1:n})} &= \frac{\int p(\boldsymbol{x}_{1:n}|\boldsymbol{\theta}, M_1)p(\boldsymbol{\theta}, M_1)d\boldsymbol{\theta}}{p(\boldsymbol{x}_{1:n}|M_0)} \frac{p(M_1)}{p(M_0)}, \\
&= \frac{p(\boldsymbol{x}_{1:n}|M_1)}{p(\boldsymbol{x}_{1:n}|M_0)} \frac{p(M_1)}{p(M_0)}, \\
&= \frac{\prod_{i=1}^n p(\boldsymbol{x}_i|\boldsymbol{x}_{1:i-1}|M_1)}{\prod_{i=1}^n p(\boldsymbol{x}_i|\boldsymbol{x}_{1:i-1}|M_0)} \frac{p(M_1)}{p(M_0)}, \\
&= \frac{p(\boldsymbol{x}_n|\boldsymbol{x}_{1:n-1}, M_1)}{p(\boldsymbol{x}_n|\boldsymbol{x}_{1:n-1}, M_0)} \frac{p(M_1|\boldsymbol{x}_{1:n-1})}{p(M_0|\boldsymbol{x}_{1:n-1})}, \\
&= \frac{\int p(\boldsymbol{x}_n|\boldsymbol{\theta}, \boldsymbol{x}_{1:n-1}, M_1)p(\boldsymbol{\theta}|\boldsymbol{x}_{1:n-1}, M_1)d\boldsymbol{\theta}}{p(\boldsymbol{x}_n|\boldsymbol{x}_{1:n-1}, M_0)} \frac{p(M_1|\boldsymbol{x}_{1:n-1})}{p(M_0|\boldsymbol{x}_{1:n-1})},
\end{aligned}$$

where the last expression stresses the recursive definition of the posterior odds factor in terms of products of posterior predictive densities. The posterior distribution of $\boldsymbol{\theta}|\boldsymbol{x}_{1:n}, M_1 \sim \text{Dirichlet}(\boldsymbol{\alpha}_n)$ where $\boldsymbol{\alpha}_n = \boldsymbol{\alpha}_{n-1} + \boldsymbol{x}_n$ with $\boldsymbol{\alpha}_0$ the initial prior parameter choice. The posterior predictive densities are easily computed as

$$p(\boldsymbol{x}_n|\boldsymbol{x}_{1:n-1}, M_1) = \frac{\Gamma(\sum_i x_{n,i} + 1)}{\prod_i \Gamma(x_{n,i} + 1)} \frac{\Gamma(\sum_i \alpha_{n-1,i})}{\prod_i \Gamma(\alpha_{n-1,i})} \frac{\prod_i \Gamma(\alpha_{n-1,i} + x_{n,i})}{\Gamma(\sum_i \alpha_{n-1,i} + x_{n,i})}$$

and

$$p(\boldsymbol{x}_n|\boldsymbol{x}_{1:n-1}, M_0) = \frac{\Gamma(\sum_i x_{n,i} + 1)}{\prod_i \Gamma(x_{n,i} + 1)} \prod_i \theta_{0,i}^{x_{n,i}}.$$

It follows that

$$\frac{p(M_1|\boldsymbol{x}_{1:n})}{p(M_0|\boldsymbol{x}_{1:n})} = \frac{\Gamma(\sum_i \alpha_{n-1,i})}{\Gamma(\sum_i \alpha_{n-1,i} + x_{n,i})} \frac{\prod_i \Gamma(\alpha_{n-1,i} + x_{n,i})}{\prod_i \Gamma(\alpha_{n-1,i})} \frac{1}{\prod_i \theta_{0,i}^{x_{n,i}}} \frac{p(M_1|\boldsymbol{x}_{1:n-1})}{p(M_0|\boldsymbol{x}_{1:n-1})},$$

(21)

(22)

where

$$\boldsymbol{\alpha}_n = \boldsymbol{\alpha}_{n-1} + \boldsymbol{x}_n.$$

(23)

To see the alternative "betting" form of the posterior odds when $\alpha_{0,i}$ are integer valued, note that $x_{n,i}$ is equal to 1 for a single index $i \in \{1, \ldots, d\}$ and is equal to 0 for the remaining indices. The previous expression can then be written

$$\frac{p(M_1|\boldsymbol{x}_{1:n})}{p(M_0|\boldsymbol{x}_{1:n})} = \frac{\Gamma(\sum_i \alpha_{n-1,i})}{\Gamma(1 + \sum_i \alpha_{n-1,i})} \prod_{i=1}^d \left( \frac{\Gamma(\alpha_{n-1,i} + 1)}{\Gamma(\alpha_{n-1,i})} \right)^{x_{n,i}} \frac{1}{\prod_i \theta_{0,i}^{x_{n,i}}} \frac{p(M_1|\boldsymbol{x}_{1:n-1})}{p(M_0|\boldsymbol{x}_{1:n-1})},$$

(24)

$$= \frac{1}{\sum_i \alpha_{n-1,i}} \prod_{i=1}^d \alpha_{n-1,i}^{x_{n,i}} \frac{1}{\prod_i \theta_{0,i}^{x_{n,i}}} \frac{p(M_1|\boldsymbol{x}_{1:n-1})}{p(M_0|\boldsymbol{x}_{1:n-1})},$$

(25)

$$= \prod_{i=1}^d \left( \frac{\alpha_{n-1,i}}{\sum_j \alpha_{n-1,j}} \right)^{x_{n,i}} \frac{1}{\prod_i \theta_{0,i}^{x_{n,i}}} \frac{p(M_1|\boldsymbol{x}_{1:n-1})}{p(M_0|\boldsymbol{x}_{1:n-1})},$$

(26)

(27)

where the first term can be recognized as the mean of a Dirichlet distribution with parameter $\boldsymbol{\alpha}_{n-1}$. Tidying up terms yields

$$\frac{p(M_1|\boldsymbol{x}_{1:n})}{p(M_0|\boldsymbol{x}_{1:n})} = \prod_{i=1}^{d} \left( \frac{\alpha_{n-1,i}}{\theta_{0,i} \sum_j \alpha_{n-1,j}} \right)^{x_{n,i}} \frac{p(M_1|\boldsymbol{x}_{1:n-1})}{p(M_0|\boldsymbol{x}_{1:n-1})}, \tag{28}$$

$$\tag{29}$$

## A.6 Proof of Theorem 2.1 (Martingale Property of Posterior Odds)

*Proof.*

$$\begin{aligned}
E_{M_0}[O_{n+1}(\boldsymbol{\theta}_0)|\mathcal{F}_n] &= \int \frac{p(\boldsymbol{x}_{n+1}|\boldsymbol{x}_{1:n}, M_1)}{p(\boldsymbol{x}_{n+1}|\boldsymbol{x}_{1:n}, M_0)} O_n(\boldsymbol{\theta}_0) p(\boldsymbol{x}_{n+1}|\boldsymbol{x}_{1:n}, M_0) d\boldsymbol{x}_{n+1} \\
&= O_n(\boldsymbol{\theta}_0) \int p(\boldsymbol{x}_{n+1}|\boldsymbol{x}_{1:n}, M_1) d\boldsymbol{x}_{n+1} \\
&= O_n(\boldsymbol{\theta}_0),
\end{aligned}$$

where $\mathcal{F}_n = \sigma(\boldsymbol{x}_1, \boldsymbol{x}_2, \dots, \boldsymbol{x}_n)$. $\square$

## A.7 Proof of Theorem 2.2 (Construction of Test-Martingale)

First, the following lemma is required

**Lemma A.1.** *(Ville's Maximal Inequality) If $Z_n$ is a nonnegative supermartingale with respect to the filtration $\mathcal{F}_n$, then*

$$\mathbb{P}[\exists n \in \mathbb{N}_0 : Z_n \geq u] \leq \frac{Z_0}{u} \tag{30}$$

*Proof.* See Ville [1939], Howard et al. [2020] $\square$

Theorem 2.1 shows that $O_n(\boldsymbol{\theta}_0)$ is a nonnegative martingale (and therefore also a supermartingale) under the null hypothesis with initial value $O_0(\boldsymbol{\theta}_0) = 1$. The result then follows immediately from Lemma A.1.

## A.8 Proof of Theorem 2.3 (Asymptotic Properties of Bayes Factors)

From Equation (4),

$$\begin{aligned}
\log O_n(\boldsymbol{\theta}_0) &= \log \text{Beta}(\boldsymbol{\alpha}_0 + \boldsymbol{S}_n) - \log \text{Beta}(\boldsymbol{\alpha}_0) - \sum_i S_i^n \log \boldsymbol{\theta}_{0,i} \\
&= \sum_i \log \Gamma(\alpha_{0,i} + S_i^n) - \log \Gamma(|\boldsymbol{\alpha}_0 + \boldsymbol{S}^n|) + \\
&\quad \log \Gamma(|\boldsymbol{\alpha}_0|) - \sum_i \log \Gamma(\alpha_{0,i}) \\
&\quad - \sum_i S_i^n \log \boldsymbol{\theta}_{0,i}.
\end{aligned} \tag{31}$$

Using Stirlings approximation $\log \Gamma(z) = z \log z - z + o(\log z)$

$$
\begin{aligned}
\log O_n(\boldsymbol{\theta}_0) &= \sum_i (\alpha_{0,i} + S_i^n) \log(\alpha_{0,i} + S_i^n) - (\alpha_{0,i} + S_i^n) \\
&\quad - (|\boldsymbol{\alpha}_0| + n) \log(|\boldsymbol{\alpha}_0| + n) + (|\boldsymbol{\alpha}_0| + n) + \\
&\quad - \sum_i S_i^n \log \theta_{0,i} + o(\log n) \\
&= \sum_i (\alpha_{0,i} + S_i^n) \log \left( \frac{\alpha_{0,i} + S_i^n}{(|\boldsymbol{\alpha}_0| + n)} \right) \\
&\quad - \sum_i S_i^n \log \theta_{0,i} + o(\log n) \\
&= \sum_i S_i^n \log \left( \frac{\alpha_{0,i} + S_i^n}{(|\boldsymbol{\alpha}_0| + n)} \frac{1}{\theta_{0,i}} \right) + o(\log n) \\
\frac{1}{n} \log O_n(\boldsymbol{\theta}_0) &= \sum_i \frac{S_i^n}{n} \log \left( \frac{\alpha_{0,i} + S_i^n}{(|\boldsymbol{\alpha}_0| + n)} \frac{1}{\theta_{0,i}} \right) + o\left( \frac{\log n}{n} \right).
\end{aligned}
\tag{32}
$$

$\frac{S_i^n}{n}$ and $\frac{\alpha_{0,i}+S_i^n}{(|\boldsymbol{\alpha}_0|+n)}$ converge to $\theta_i$ almost surely by the strong law of large numbers. It follows that

$$
\frac{1}{n} \log O_n(\boldsymbol{\theta}_0) \overset{\text{a.s.}}{\to} \sum_i \theta_i \log \left( \frac{\theta_i}{\theta_{0,i}} \right),
\tag{33}
$$

by Slutsky's theorem and the continuous mapping theorem, which can be recognized as the Kullback-Leibler divergence of a $\mathrm{Multinomial}(1, \boldsymbol{\theta})$ distribution from a $\mathrm{Multinomial}(1, \boldsymbol{\theta}_0)$ distribution.

### A.9 Type I Error Probability Simulation

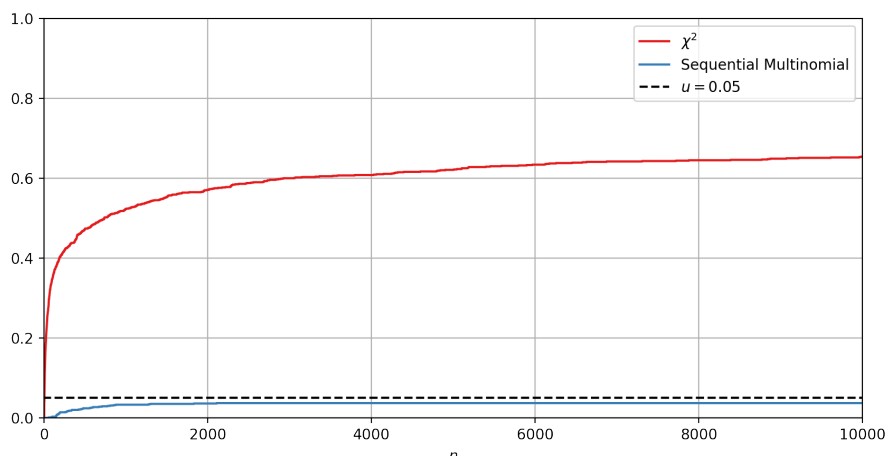

Figure 4: Estimated probability of falsely rejecting the null by sample size $n$ under stopping rules (red) when the $\chi^2$ $p$-value falls below $0.05$ (blue) when the sequential $p$-value from Equation (8) falls below $0.05$. Estimates based on 10000 simulations. Null rejected incorrectly 654 and 37 times by $\chi^2$ and sequential multinomial tests respectively.

### A.10 Proof of Theorem 4.1

The following two lemmas are required prove Theorem 4.1. Consider a poisson point process the intensity function $\lambda(t)$. The probability density over the event time $t_i$ conditional on the previous

event having arrived at time $t_{i-1}$ is given by

$$p(t_i|t_{i-1}) = \lambda(t_i)e^{-\int_{t_{i-1}}^{t_i} \lambda(s)ds}1_{(t_{i-1},\infty]}(t_i). \tag{34}$$

This is used to prove the following lemma

**Lemma A.2.** *Let $t_{i-1}$ denote the time of the previously observed event. Suppose the current time is $T > t_{i-1}$. The probability density over the next event time $t_i$ conditional on no observation having occured in $(t_{i-1}, T]$ is given by*

$$p(t_i|t_i > T) = \lambda(t_i)e^{-\int_T^{t_i} \lambda(s)ds}1_{(T,\infty]}(t_i). \tag{35}$$

*Proof.* The probability of no event taking place $(t_{i-1}, T]$ is given by the Poisson$(\Lambda(t_{i-1}, T])$ distribution

$$\mathbb{P}[N(t_{i-1}, T] = 0] = e^{-\int_{t_{i-1}}^T \lambda(s)ds}. \tag{36}$$

The conditional density for $t_i$ given $t_i > T$ is obtained by conditioning on $N(t_{i-1}, T] = 0$ as follows

$$\begin{aligned}
p(t_i|t_i > T) &= \frac{\lambda(t_i)e^{-\int_{t_{i-1}}^{t_i} \lambda(s)ds}1_{(T,\infty]}(t_i)}{e^{-\int_{t_{i-1}}^T \lambda(s)ds}} \\
&= \lambda(t_i)e^{-\int_T^{t_i} \lambda(s)ds}1_{(T,\infty]}(t_i).
\end{aligned} \tag{37}$$

$\square$

The following lemma asks, given two inhomogeneous Poisson point processes 0 and 1, what is the probability that the next event comes from 0?

**Lemma A.3.** *Consider two inhomogeneous Poisson point processes with intensities $\lambda_0(t) = e^{\delta_0}\lambda(t)$ and $\lambda_1 = e^{\delta_1}\lambda(t)$. Let the current time be denoted $T$. The probability that the next event is from process 1 is given by*

$$\frac{e^{\delta_1}}{e^{\delta_0} + e^{\delta_1}}. \tag{38}$$

*Proof.* Let $\tau_0$ and $\tau_1$ denote the next event times from process 0 and 1 respectively. From Lemma A.2

$$p(\tau_1|\tau_1 > T) = \lambda_1(\tau_1)e^{-\Lambda_1(T,\tau_1]}1_{(T,\infty]}(\tau_1)$$
$$p(\tau_0|\tau_0 > T) = \lambda_0(\tau_0)e^{-\Lambda_0(T,\tau_0]}1_{(T,\infty]}(\tau_0),$$

where $\Lambda_i(T, \tau_i] = \int_T^{\tau_i} \lambda_i(s)ds$ The probability that the next event is from process 1 is given by

$$\begin{aligned}
\mathbb{P}[\tau_1 < \tau_0|\tau_0, \tau_1 > T] &= \int_T^\infty \lambda_0(\tau_0)e^{-\Lambda_0(T,\tau_0]} \int_T^{\tau_0} \lambda_1(\tau_1)e^{-\Lambda_1(T,\tau_1]}d\tau_1 d\tau_0 \\
&= \int_T^\infty \lambda_0(\tau_0)e^{-\Lambda_0(T,\tau_0]} \left(1 - e^{-\Lambda_1(T,\tau_0]}\right) d\tau_0 \\
&= 1 - \int_T^\infty \lambda_0(\tau_0)e^{-\int_T^{\tau_0} \lambda_0(s)+\lambda_1(s)}d\tau_0 \\
&= 1 - \frac{e^{\delta_0}}{(e^{\delta_0} + e^{\delta_1})} \int_T^\infty (e^{\delta_0} + e^{\delta_1})\lambda(\tau_0)e^{-(e^{\delta_0}+e^{\delta_1})\int_T^{\tau_0} \lambda(s)}d\tau_0 \\
&= 1 - \frac{e^{\delta_0}}{(e^{\delta_0} + e^{\delta_1})} \\
&= \frac{e^{\delta_1}}{(e^{\delta_0} + e^{\delta_1})}.
\end{aligned}$$

$\square$

The proof of Theorem 4.1 can now be given using Lemma A.3 and the superposition property [Kingman, 1992] of Poisson processes.

*Proof.* Consider $d$ inhomogeneous Poisson point processes with intensity functions $\lambda_i(t) = \rho_i e^{\delta_i} \lambda(t)$ for $i \in \{1, 2, \ldots, d\}$. Choose any process of interest $j$, with corresponding intensity $\rho_j e^{\delta_j} \lambda(t)$. Let the union of all timestamps from the other $i \neq j$ processes be combined into a new "not $j$" process. By the superposition property of the Poisson process, the combined timestamps form a new Poisson process with intensity function $\lambda_u(t) = \sum_{i \neq j} \rho_i e^{\delta_i} \lambda(t)$. This reduces the problem to a comparison of two inhomogeneous Poisson point processes. From Lemma A.3, the probability that the next event corresponds to process $j$ is then

$$\frac{\rho_j e^{\delta_j}}{\rho_j e^{\delta_j} + \sum_{i \neq j} \rho_i e^{\delta_i}} = \frac{\rho_j e^{\delta_j}}{\sum_i \rho_i e^{\delta_i}}. \tag{39}$$

$\square$

### A.11 Simulation: Time Inhomogeneous Bernoulli Processes

Consider the following example. An experimental unit $i$ is randomly assigned to one of 3 arms with probabilities $\boldsymbol{\rho} = (0.1, 0.3, 0.6)$. Let $g(i)$ map the unit to the arm index to which it is assigned. A Bernoulli success is observed for unit $i$ with probability $e^{\mu(i)} e^{\delta_{g(i)}}$ with $\boldsymbol{\delta} = (\log 0.2, \log 0.3, \log 0.4)$ and $\mu(i) = \frac{1}{2} \sin(\frac{7\pi i}{n}) + \frac{1}{2}$. Confidence sequences for contrasts $\delta_2 - \delta_1$ and $\delta_2 - \delta_0$, obtained through Equation (15), are shown in Figure 5. The sequential p-value for testing the hypothesis $\delta_0 - \delta_1 \geq 0$ and $\delta_0 - \delta_2 \geq 0$, obtained through Equation (16), are shown using the right axis of Figure 5. The p-value is less than $u = 0.05$ for all $n \geq 573$. This is the smallest $n$ for which the joint confidence set over these contrasts fails to intersect with the set defined by the hypothesis (the lower left quadrant) as shown in Figure 6.

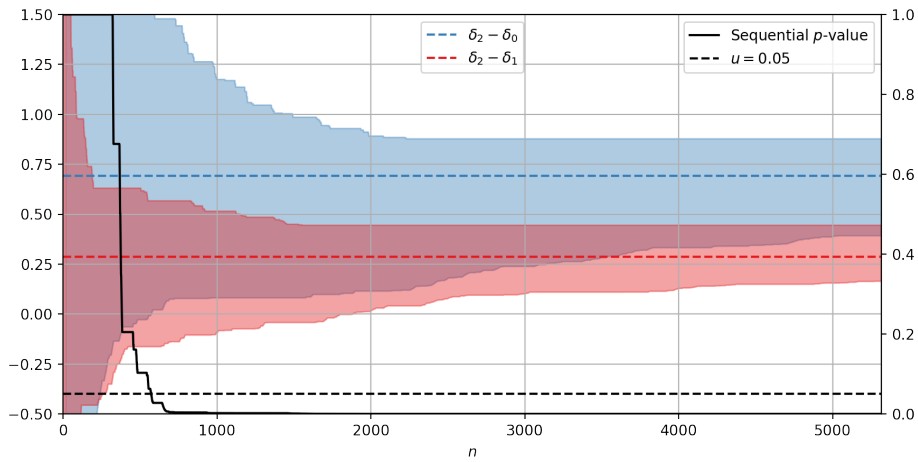

Figure 5: (Left axis) 0.95 Simultaneous confidence sequences for $\delta_2 - \delta_1 = \log(0.4) - \log(0.3) \approx 0.29$ and $\delta_2 - \delta_0 = \log(0.4) - \log(0.2) \approx 0.69$ provided by Corollary 3.2 and obtained via the solution to Equation (15). The confidence sequences for $\delta_2 - \delta_0$ and $\delta_2 - \delta_1$ are completely positive for $n \geq 573$ and $n \geq 1882$ respectively, after which one can conclude with probability $1 - u$ that arm 2 is optimal. (Right axis) Sequential p-value for testing the null hypothesis $\delta_0 \geq \delta_1$ and $\delta_0 \geq \delta_2$ obtained by solving Equation (16). The p-value is less than critical value $u = 0.05$ for all $n \geq 573$.

### A.12 Simulation: Time Inhomogeneous Poisson Processes

Consider the following example with only two arms, such as a canary test designed to test if a new software version produces more errors. Units are assigned to arms with probability $\boldsymbol{\rho} = (0.8, 0.2)$. $\lambda_1(t)$ can be expressed in terms of $\lambda_0(t)$ as $\lambda_1(t) = \rho e^{\delta} \lambda_0(t)$ with $\rho = \frac{\rho_1}{\rho_0}$ and $e^{\delta} = e^{\delta_1 - \delta_0}$. Let $\delta = 1.5$ and $\lambda_0(t) = 2000 \text{sigmoid}(\sin(10\pi t) + 8t - 4)$. Point process realizations are obtained by

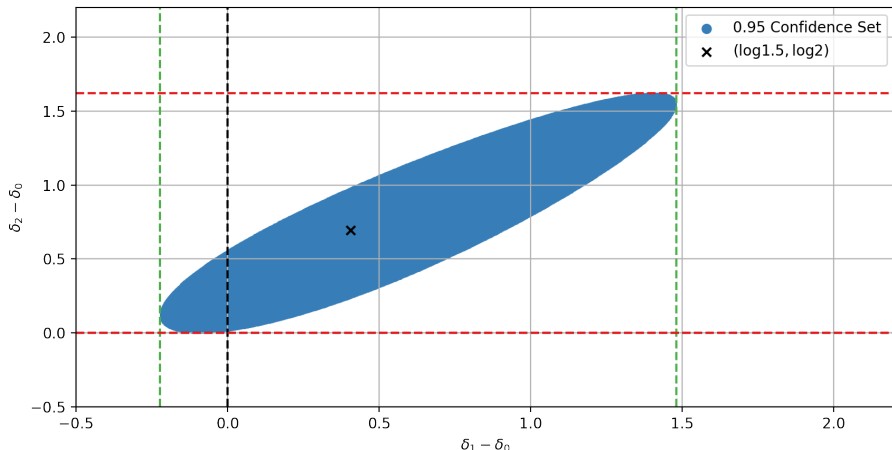

Figure 6: 0.95 joint confidence set for $\delta_2 - \delta_0$ and $\delta_1 - \delta_0$ at $n = 573$. (Black cross) True parameter values $(\log 3/2, \log 2)$. (Red dashed) $l^-_{573,\boldsymbol{b}}$ and $l^+_{573,\boldsymbol{b}}$, (Green dashed) $l^-_{573,\boldsymbol{c}}$ and $l^+_{573,\boldsymbol{c}}$ as in Corollary 3.2 with $\boldsymbol{b} = (-1, 0, 1)$ and $\boldsymbol{c} = (0, -1, 1)$.

thinning a homogeneous Poisson point process with rate 2000 [Lewis and Shedler, 1979]. Figure 7 shows the point process realizations, intensities, and counting processes for each arm. Figure 8 shows the confidence sequence for $\delta$ and the sequential $p$-value for testing equality $\delta = 0$ ($\boldsymbol{\theta} = \boldsymbol{\rho}$).

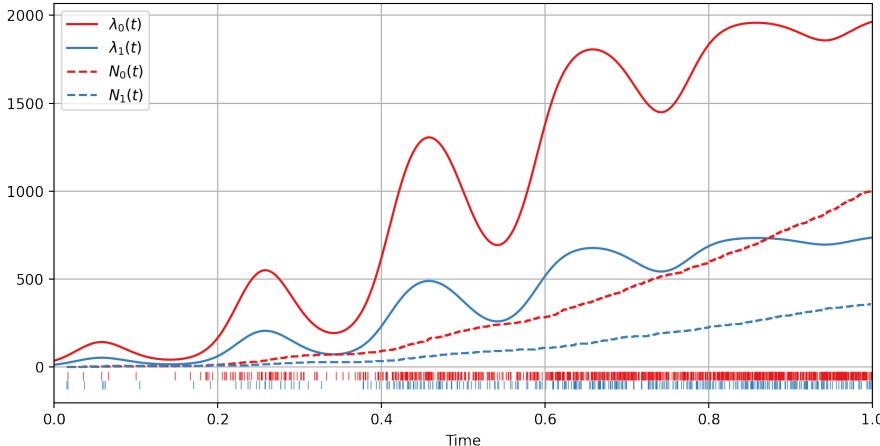

Figure 7: Inhomogeneous Poisson point process intensities $\lambda_0(t) = 2000\text{sigmoid}(\sin(10\pi t) + 8t - 4)$ and $\lambda_1(t) = \frac{1}{4}e^{\frac{3}{2}}\lambda_0(t)$. Associated counting processes $N_0(t)$ and $N_1(t)$. Point process realizations (rug-plots).

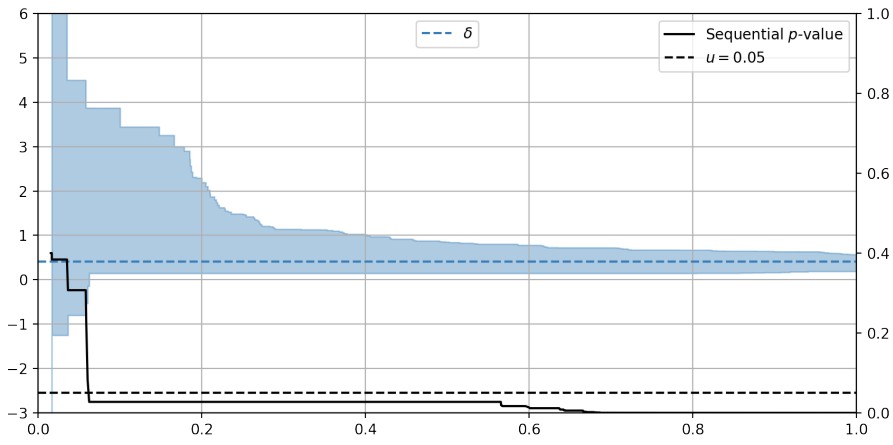

Figure 8: (Left axis) 0.95 *continuous-time* Confidence sequence for $\delta = 1.5$. (Right axis) Sequential *p*-value for testing equality i.e. $\delta = 0 \Rightarrow \boldsymbol{\theta} = \boldsymbol{\rho}$.