# OpenReview forum: "Anytime-Valid Inference For Multinomial Count Data"
_NeurIPS.cc/2022/Conference — NeurIPS 2022 Accept_

### Official Review · Reviewer_FAoW · 2022-07-07

**Rating:** 5
**Confidence:** 3
**Soundness:** 3 good
**Presentation:** 3 good
**Contribution:** 2 fair

**Summary:**

The paper introduces a sequential test for multinomial hypotheses using a martingale construction. It then uses this result to develop a sequential test of equality and contrasts in inhomogeneous Bernoulli processes and time-inhomogeneous Poisson counting processes.

**Questions:**

Why do you assume  "departures from the null to be small "? Do the results hold even when this assumption is not true?

Why do you focus on Byes factors and then derive results  about frequentist Type I and II errors? Then why did you not consider a likelihood ratio test instead of the Bayes factor ?.

**Limitations:**

The assumption "departures from the null to be small" is very strong.

**Strengths And Weaknesses:**

The strengths are (1) the authors use martingale inequalities to construct a test martingale that
controls the frequentist Type I probability below a desired level u. (2) they obtain a confidence with a coverage guarantee of at least
1-u.

The main assumption of the test is that "departures from the null to be small and encode this information into the Dirichlet prior". Null-hypotheses are never fully true and, therefore, the above assumption decreases the robustness of the test.
I am not sure the result is really novel and in any case it seems to be an application of the results in Test Martingales, Bayes Factors
and p-Values by Shafer et al.. I believe a statistical conference/journal would be a more appropriate venue for this paper, where the contributions of this paper can be discussed with more details with respect to the state of the art.
It is also not clear to me why the authors focus on Bayes factors and then derive results about frequentist Type I and II errors. Then why did they not consider a likelihood ratio test instead of a Bayes factor ? From a Neurips perspective, the authors are considering a simple problem (count data) and other approaches could be used. For instance, one could simply compute the posterior of theta and then use alpha-% credible intervals for drawing conclusions about theta_0. This approach would work quite well in practice considering the goal is to estimate theta from counts.

---

> ### Author Response · Authors · 2022-07-31
> **Some misunderstandings of our theoretical results and potentially missed details in our appendix. Some of the requested comparisons and discussions are already in the supplemental material.**
>
> We would like to thank the reviewer for reading our paper and are grateful for the opportunity to clarify a misunderstanding which we feel has led to unfair criticism of our work. We would appreciate the reviewer reading our rebuttal and reevaluating our paper with a corrected understanding of the applicability of our results.
>
> The primary criticism is *"The assumption "departures from the null to be small" is very strong"*.  Our results, to the contrary, do not depend on this at all. This assumption is not made in any of our theorems. In particular, we do not assume that the parameter is a random variable with a Dirichlet distribution. In fact, we do not even assume that the parameter is random. We consider the parameter to be some fixed unknown as a frequentist would. We integrate the likelihood ratio with respect to a Dirichlet weight function simply to obtain a test statistic. We then study the *frequentist behaviour* of this test statistic under *frequentist assumptions*.
>
> - Type-I error is controlled under the null hypothesis under continuous monitoring through a time uniform bound
> - The test is consistent. It is asymptotically power 1 for all $\theta \neq \theta_0$. This means the null is rejected almost surely for all alternative parameter values.
> The confidence sequence covers the true $\theta$ at all times with probability $>1-u$.
> - None of these results require the parameter to be considered a random variable with a Dirichlet Distribution. We simply consider it a fixed unknown.
>
> The weight or mixture distribution (we used a Dirichlet) should be considered by frequentists simply as a tool to obtain a test statistic that is a Martingale under the null hypothesis (a test Martingale). For details on this construction see Kaufmann and Koolen 2021. Guidance on choosing a mixture distribution can be found in section 5.2 “choosing the mixture distribution” of Johari et al 2017. Examples of this construction can be found in section 3.2 “conjugate mixture boundaries” of Howard et al 2021.
>
> The mixture distribution does not affect the Type I and consistency guarantees but can influence how quickly the sequential test rejects the null. Qualitatively, smaller(larger) departures from the null are rejected quicker when choosing a mixture distribution that puts more(less) mass close to the null. We choose a Dirichlet that concentrates mass close to the null because we observe small effect sizes in practice in our applications. The Dirichlet concentration parameter is, however, a configurable input to our procedure. If one expects larger effect sizes in their own applications, then our proposal is just as valid by choosing a more diffuse Dirichlet distribution, or even a uniform distribution over the simplex (a special case of a Dirichlet). Johari et al. 2017 go into further detail, but the theoretically optimal average stopping time is obtained by matching the mixture distribution to the distribution of treatment effects observed in practice.
>
> We acknowledge that there is debate in the statistics community around null-hypothesis significance testing, but the majority of statisticians accept this formulation and would be interested in our results. Sequential Tests of point-nulls lead to confidence sequences. Confidence sequences and time uniform bounds are fundamental building blocks in many multi-armed bandit algorithms such as the LUCB algorithm, which are used by the machine learning community for best arm identification.
>
> Our novel contributions are sequential tests of inhomogeneous Bernoulli and Poisson counting processes, which are important applications in industry. *Ours is the first approach to possess the desired anytime-valid time-uniform guarantees*, which are essential for performing sequential inference on real-time streams of data. We discuss the deficiencies in existing tests, including the requested *likelihood ratio test*, in appendix section A.1 “Limitations of Fixed-n Testing”, and motivate our proposal in section A.2 “Solutions via Sequential Testing”. We then demonstrate a direct comparison through simulation of our proposed test to the classical counterpart - the *Chi-squared test*. This is found in appendix section A.9 Figure 4. The problem with the likelihood ratio test and other approaches are that they do not control Type-I errors under continuous monitoring. Similarly, classical confidence intervals and the proposed *Bayesian credible interval* do not cover the true parameter *for all times* - only at a *fixed-n* time.
>
> It is possible to write this paper without any references to Bayes. We included Bayesian terminology in the hope that this could appeal to both Bayesian and frequentists. Our proposal uses statistics that are fundamental to Bayesian analysis, but provides frequentist guarantees, and hope it is attractive to both. The mixture likelihood ratio statistic used by frequentists can be interpreted by Bayesians as a Bayes factor and vice versa, Berger 1999.

---

> > ### Comment · Reviewer_FAoW · 2022-08-09
> > **Reply to authors**
> >
> > Thank you for your answers. I have only two further comments.
> >
> > 1)  I don't understand your answer "we consider the parameter to be some fixed unknown as a frequentist would. We integrate the likelihood ratio with respect to a Dirichlet weight function simply to obtain a test statistic"
> > You are computing a BF. In M0 you are placing an atomic measure prior on theta_0 (equation (1)) and in M1 you are placing a Dirichlet prior on theta_0 to account for the alternative hypothesis  (equation (2)), that is theta different from theta_0. This is standard in BFs. Then you provide results on the sequence of BFs, I mean Th.2.1 etc.
> >
> > 2) About you explanation for alpha_0 = k Theta_0, I suggest to add it to the paper.

---

> > > ### Author Response · Authors · 2022-08-09
> > > **Replies on the use of Bayes factors**
> > >
> > > It is usually observed that Bayesian procedures perform well under optional stopping (see [1] for example), yet we seek a sequential test that has good frequentist properties (controls Type I error for all parameter values, consistent for all parameter values). For instance, a frequentist might be very concerned with two parts of the Bayesian procedure
> > >
> > > - How do your prior odds impact the analysis?
> > > - How does your prior over the unknown parameters the analysis? What guarantees do I get from this procedure?
> > >
> > > A frequentist would seek Type-I error control for all parameter values, not Type-I error on average with respect to priors.
> > > We take inspiration from Bayesian methods by looking at the Bayes factor, but then we study the frequentist properties of this object. You can consider us adding frequentist guarantees to this Bayesian procedure. Once we compute the Bayes factor, we cease to use it as a Bayesian would, and study it through frequentist lenses instead. For instance, a Bayesian would use the Bayes factor to compute posterior probabilities of the alternative and null hypothesis and show credible intervals. We don't go in this direction. Instead, we recognize that this object is a nonnegative supermartingale under the null hypothesis, and use Martingale inequalities (Ville's) to bound the probability that this object ever gets large and consequently control Type-I error under the proposed stopping rule.
> > >
> > > Instead of credible intervals, which only have the desired $1-\alpha$ coverage at a fixed $n$ and only if the multinomial parameter is random with a Dirichlet distribution, we present confidence sequences, which are very different objects - they cover the multinomial parameter $\theta$ whatever value it takes (doesn't require the random dirichlet assumption, doesn't even require it to be random, i.e. this is the frequentist perspective) and it holds for all $n$.
> > >
> > > In short, we have taken inspiration from Bayesian methods and considered the Bayes factor as an interesting mathematical object. We have then constructed a frequentist sequential test that utilizes this object.
> > >
> > > We have nothing against Bayesian methods. Indeed, when prior information is available we would highly recommend a pure Bayesian analysis. Sometimes, however, we recognize that statisticians seek frequentist properties. Such cases arise when an experiment is perhaps the very first in a new application area, for which there is no prior knowledge with which to base priors.
> > >
> > > An alternative perspective is that we could have written this paper without any references to Bayes. The Bayes factor appears in the frequentist literature on sequential testing also, except they don't call it a Bayes factor, they call it a likelihood ratio mixture. Such likelihood ratio mixtures were studied by Abraham Wald and made no reference to Bayes [2].
> > >
> > > if the Bayesian terminology has led to confusion, we would be happy to entertain removing all Bayesian verbiage in a final version of the paper, replacing "Bayes factor" by "likelihood ratio mixture", and replacing "posterior odds" with "martingale process". We thought it might be nice to incorporate some Bayesian nomenclature to highlight the connection to Bayesian analysis, but not at the cost of clarity. Please let us know what you would prefer.
> > >
> > > [1] Rouder JN. Optional stopping: no problem for Bayesians. Psychon Bull Rev. 2014 Apr;21(2):301-8. doi: 10.3758/s13423-014-0595-4. PMID: 24659049.
> > > [2] Wald, Abraham. 1947. Sequential analysis. New York: J. Wiley & Sons.

---

### Official Review · Reviewer_PFqX · 2022-07-11

**Rating:** 8
**Confidence:** 4
**Soundness:** 4 excellent
**Presentation:** 3 good
**Contribution:** 4 excellent

**Summary:**

This paper proposes new sequential hypothesis tests and corresponding confidence sets and intervals for multinomial data. These tests provide "always-valid" guarantees, i.e. controlled Type I error probability under optional stopping/continuation.
First, the authors consider the direct application of detecting departures from a specified multinomial parameter, that is sample ratio mismatch tests.
Second, they consider the problem of comparing the success rate of k arms corresponding to inhomogeneous Bernoulli processes. The parameters of the Bernoullis can vary with time (but in the same way for all arms).
This problem is mapped to a multinomial one by considering the probability of each arm producing the next success conditioned on the event that there is a success.
This point of view has several advantages:
1. It allows using the sequential constructions for tests, confidence sets and intervals proposed in the first part.
2. It only needs to count success observations.
3. The time varying effect gets canceled out.

Finally, the problem of comparing the intensity of inhomogeneous Poisson counting processes is also mapped to a multinomial problem by showing that the probability of the next observation belonging to the i-th process is a multinomial random variable.


**Questions:**

The test of [1, Section 6.2] (based on [2])  is also based on a sequential test for multinomial count data, with the difference that the multinomial count data is conditioned on a continuous variable, making the alternative hypothesis composite and allowing to build a nonparametric two-sample test. I think it is worth mentioning it since it has a similar spirit and is based on the same tools/properties (likelihood ratio, multinomial model, sequential Bayesian updating, nonnegative supermartingales, optional stop/continuation).

*[1] Lhéritier, A., & Cazals, F. (2019). Low-complexity nonparametric Bayesian online prediction with universal guarantees. Advances in Neural Information Processing Systems, 32.*

*[2] Lhéritier, A., & Cazals, F. (2018). A sequential non-parametric multivariate two-sample test. IEEE Transactions on Information Theory, 64(5), 3361-3370.*

Minor comments:

Line 49: $\mathbf{x}_1,\mathbf{x}_2,\mathbf{x}_3,…$ are vectors so I guess they are dummy coded variables, maybe it is worth mentioning it.

Line 176: maybe you could say "so that the time-varying effect $\mu(t)$" to explicitly introduce $\mu(t)$

Eq just after line 196: $\mathbf{a}$ should be in bold font

References are not sorted alphabetically or numerically, so it is difficult to find things in there.


**Limitations:**

Yes

**Strengths And Weaknesses:**

I enjoyed reading this paper that addresses difficult and interesting statistical problems.
Although the paper is very well written, I found it a bit unfortunate that part of the introductory material had to go to the Appendix, but I understand it was necessary for space constraints.
As explained in the paper, the construction of the sequential test for multinomial point hypotheses follows standard steps that are well known in the community and the theoretical results of this part are straightforward consequences of known results.

From my point of view, the most interesting and novel contributions are those of Section 3 and 4, i.e. how to turn the problem of comparing time-inhomogeneous Bernoulli and Poisson counting processes into a multinomial problem.
Moreover, these problems have important practical applications that are very relevant for the NeurIPS community.

---

> ### Author Response · Authors · 2022-08-02
> **Excellent review. Thoughtful comments and useful suggestions to related literature.**
>
> We are very grateful to the reviewer for such a thorough review of our work. The thoughtful comments we have received demonstrate a complete understanding of our work and the suggested papers show familiarity with the relevant literature. We very much enjoyed reading these papers. In particular, reference [2]'s construction of a p-value follows the construction in
>
> [3] Algoet, P. (1992). Universal Schemes for Prediction, Gambling and Portfolio Selection. The Annals of Probability, 20(2), 901–941. https://doi.org/10.1214/aop/1176989811
>
> We found the gambling interpretation to have a remarkable intersection with the recent work on E-values (which share bayes factor and gambling strategy interpretation)
>
> [4] Grünwald, P., de Heide, R., & Koolen, W. (2019). Safe Testing. arXiv. https://doi.org/10.48550/ARXIV.1906.07801
>
> In an earlier version of this paper, we had a proof of consistency which was very similar in approach to reference [2], using the consistency results of
>
> [5] Walker, S., Damien, P., & Lenk, P. (2004). On Priors With a Kullback–Leibler Property. Journal of the American Statistical Association, 99(466), 404–408. https://doi.org/10.1198/016214504000000386
>
> By demonstrating that the Dirichlet prior possesses the Kullback-Leibler property (defined as putting positive mass on all KL-neighbourhoods of all densities [and hence also the "true" density]), then using the results of [5] the posterior predictive density converges to the true conditional density, as the dirichlet posterior concentrates on the true $\theta$, yielding a consistent test. We feel that the connection to the referenced papers is that this yields a pointwise universal prediction scheme in our application.  We opted not to use this proof strategy, as it seemed to import too much machinery, while a much simpler proof sufficed for the scope of our paper. We look forward to exploring our results deeper through the perspectives of the papers that the reviewer referenced.
>
> We also regret that we had to move much of the literature review and motivation to the appendix. As the reviewer correctly guessed, we were struggling to make space. We would be happy to try and adjust this in a revised version of the paper.
>
> A quick note on references [1,2], their constructions for sequential two-sample tests have some very insightful ideas, but we wouldn't be able to immediately apply their methods in our applications. As we understand it, these papers deal with observations $(l_i, z_i)$ with $z_i \in \mathbb{R}^d$ a feature vector and labels $l_i$ to denote treatment group, and seek to test for differences in distributions of $z_i$'s between groups. In our applications of inhomogeneous Bernoulli and Poisson counting processes we don't have any feature vector to exploit. In the Bernoulli example, all observations for all groups are equal to 1, because we only observe the Bernoulli successes and do not observe the Bernoulli failures. In the Poisson example, all of our observations for all groups are timestamps. In both cases, we have to form a test based on the information available to us, which is simply the number of observations in each treatment group, as the individual observations do not convey any additional information.

---

> > ### Comment · Reviewer_PFqX · 2022-08-09
> > **Connection to related work**
> >
> > Thanks for your very detailed response and for carefully checking the suggested material [1,2].
> > For the consistency, I agree, of course, with keeping the proofs as simple as possible and avoiding to import unnecessary machinery.
> >
> > I agree that it is not straightforward to cast the problems you address in your work as particular cases of [1,2].
> > My comment was in the following sense: [1] belongs to the family of "Sequential Hypothesis Tests of Multinomial Count Data",  since on each cell of the (growing) feature space partition it uses a KT predictor (i.e. predictive posterior for a multinomial likelihood with Dirichlet prior). So my suggestion was just to mention this connection.

---

### Official Review · Reviewer_6qpV · 2022-07-12

**Rating:** 6
**Confidence:** 3
**Soundness:** 3 good
**Presentation:** 3 good
**Contribution:** 3 good

**Summary:**

This paper proposes a sequential multinomial test where Type I error is controlled through the martingale properties and the test is consistent. Then, the paper show how to apply the proposed test to test the data generated from the Poisson counting process and the Bernoulli process.

== After rebuttal ==\
I want to thank the authors for their efforts to clarify my misunderstanding. I have more appreciation of this work now and meanwhile still have some concerns remaining.

The pros of this work after the clarification: controlling the Type I of this proposed multinomial test naturally comes from the martingale properties of a variant of the sequential probability ratio test (SPRT), and also the consistency of the proposed test is also based on the nice consistent property of the SPRT. The Dirichlet prior with $\alpha_{0,i}=k\theta_{0,i}$ is a reasonable set-up to simulate an alternative where there is only a small departure from the null.

Some concerns after the clarification: the authors point out that the comparison with the $\chi^2$ test is shown in A9 Figure 4, but from what I read, isn't this a Type I error rate comparison? Does this work have the power (Type II) comparison between the proposed test and the $\chi^2$ test? Besides, one of the biggest benefits of a sequential test is the early (optional) stopping, but the drawback is that, with the same sample size allowed, its power will be attenuated compared with a fixed-size hypothesis testing. I tried to find through the appendix to see the experimental evidence that the stopping time increases with the increasing difficulties of the alternative hypothesis, but I didn't see that in the appendix. Lastly, a sequential test usually wouldn't enjoy too much benefit when the considered alternative hypothesis only has a minimum departure from the null, and this is the setting (small departure from the null) considered in Section 2.1. I wonder does the proposed test still can reject the alternative fast in this setting. If so, what's the intuition of that?

I would love to kindly ask the authors to remind me if I miss something here. Thank you.

=== correction ===
In the end of the comments above, I was asking can the test reject the null fast under the alternative.

===final score ===
The authors did a great job to largely relieve my concerns and help me understand their work more. Thus I am willing to increase my score to 6.

**Questions:**

Thm 2.4 is confusing. From my understanding, $\mathcal{O}$($\theta$)_n is a random variable depending on $\theta$, and data points. generated according to some distributions. It seems the authors want to define a set of $\theta$ (under the null) that would not be rejected by the test, but isn't that every $\theta$ under the null has a chance to be rejected, given the data points are randomly generated?

**Limitations:**

See above.

**Strengths And Weaknesses:**

Strengths: The design of the test is technically solid. The Type I error is controlled and the test is a consistent test.
Weakness: (1) The assumption on the parameters (line 52 to line 56) about the alternative may not be accurate, but the consistency of the test heavily depends on such an assumption. (2) The motivation for developing the test instead of using some two-sample tests such as a T-test is missing. At least the author should show the experimental comparisons with some classical A/B tests.

---

> ### Author Response · Authors · 2022-07-31
> **Some misunderstandings of our theoretical results and potentially missed details in our appendix. The requested comparisons and discussions can be found in the supplemental material.**
>
> We would like to thank the reviewer for their time and are grateful for the opportunity to clarify a large misunderstanding which has caused them to believe our assumptions are far stronger than they actually are, leading to an incorrect and unfair criticism of our work. We would appreciate the reviewer reading our rebuttal and reevaluating our paper with a corrected understanding of the applicability of our results.
>
> **Weakness (1)**
>
> This is a misunderstanding. Our results, on the contrary, do not depend on this at all. This assumption is not made in any of our theorems. In particular, we do not assume that the parameter is a random variable with a Dirichlet distribution. In fact, we do not even assume that the parameter is random. We consider the parameter to be some fixed unknown as a frequentist would. We integrate the likelihood ratio with respect to a Dirichlet weight function simply to obtain a test statistic. We then study the *frequentist behaviour* of this test statistic under *frequentist assumptions*.
>
> - Type-I error is controlled under the null hypothesis under continuous monitoring through a time uniform bound
> - The test is consistent. It is asymptotically power 1 for all $\theta \neq \theta_0$. This means the null is rejected almost surely for all alternative parameter values.
> - The confidence sequence covers the true $\theta$ at all times with probability $>1-u$.
>
> The weight or mixture distribution (we used a Dirichlet) should be considered by frequentists simply as a tool to obtain a test statistic that is a Martingale under the null hypothesis (a test Martingale). For details on this construction see Kaufmann and Koolen 2021. Guidance on choosing a mixture distribution can be found in section 5.2 “choosing the mixture distribution” of Johari et al. 2017. Examples of this construction can be found in section 3.2 “conjugate mixture boundaries” of Howard et al. 2021.
>
> The mixture distribution does not affect the Type I and consistency guarantees but can influence how quickly the sequential test rejects the null. Qualitatively, smaller(larger) departures from the null are rejected quicker when choosing a mixture distribution that puts more(less) mass close to the null. We chose a Dirichlet that concentrates mass close to the null because we observe small effect sizes in practice in our applications. The Dirichlet concentration parameter is, however,  a configurable input to our procedure. If one expects larger effect sizes in their own applications, then our proposal is just as valid by choosing a more diffuse Dirichlet distribution, or even a uniform distribution over the simplex (a special case of a Dirichlet).
> Johari et al 2017 go into detail, but the theoretically optimal average stopping time is obtained by matching the mixture distribution to the distribution of treatment effects observed in practice.
>
> **Weakness (2)**
>
> This is not true. We have included a lengthy discussion motivating our proposal and have included direct comparisons with classical A/B tests. We discuss the deficiencies in existing tests in appendix section A.1 “Limitations of Fixed-n Testing”, and motivate our proposal in section A.2 “Solutions via Sequential Testing”. We demonstrate a direct comparison through simulation of our proposed test to the classical counterpart, the Chi-squared test, found in appendix section A.9 Fig 4. This simulation clearly demonstrates that the classical test does not control Type-I errors under continuous monitoring, whereas our proposal does, as is expected from our theorems.
>
> A two-sample T-Test would not be relevant for the applications we consider. A T-Test is used to test differences-in-means hypotheses, whereas we consider hypotheses on vectors of counts. The correct counterpart would be a Chi-squared test, for which we have included discussion and comparison. If we only had two treatment groups, it may be tempting to think that we could use a two-sample T-Test to test the difference in conversion probabilities in our CRO application, but this is not the case. What makes our application different is that we do not observe Bernoulli fails, we only observe Bernoulli successes. That is, observations from treatment groups are only 1’s, there are no 0’s. This is what rules out the ability to use a two-sample T-Test and requires us to create a test based purely on the counts of successes instead.
>
> Answer: $O_n(\theta_0)$ is the test-martingale at time n for testing the specific null hypothesis $\theta=\theta_0$. That is, the test-statistic is indexed by the corresponding point-null hypothesis it is testing. This notation is also used in Johari et al. 2017  (eq 8). The motivation for this notation is that we can obtain a confidence sequence by exploiting a fundamental duality between confidence intervals and p-values. We find all the null hypotheses that would not have been rejected by this data to obtain the confidence interval, which is the confidence set we express in theorem 2.4.

---

> > ### Author Response · Authors · 2022-08-08
> > **Discussion on sequential vs fixed-n tests**
> >
> > "**and this is the setting (small departure from the null) considered in Section 2.1.**" our proposed sequential test is not limited to parameterizations of the form $\alpha_{0,i} = k\theta_{0,i}$, for $k$ large. For instance, one can perform this test with $\alpha_{0,i} = 1$, yielding a uniform prior over the simplex. This means the utility of our proposal is not limited to small departures from the null at all.
> > $\alpha_{0,i}$ is a parameter that the user is free to configure, and should choose it so as to optimize the performance of their test for the effect sizes they expect. In our applications, we observe small effect sizes, so we choose $\alpha_{0,i} = k\theta_{0,i}$. If you, however, observe large departures from the null in your applications, then our proposal works just as well, and we would recommend the choice $\alpha_{0.i} = 1$ or $\alpha_{0,i} = k\theta_{0,i}$ for $k$ small to optimize the (random) stopping time. Guidance on how to choose this parameter can be found in  section 5.2 “choosing the mixture distribution”
> >
> > *[1] Johari et al. Peeking at a/b tests: Why it matters, and what to do about it. KDD ’17 https://doi.org/10.1145/3097983.3097992*
> >
> > "**Does this work have the power (Type II) comparison between the proposed test and the $\chi^2$ test?**"
> > We have not included a power comparison for the following reason. It is a fundamental result that *all sequential tests have less power than a fixed-n test as a fixed-n*. Confidence intervals at any fixed n are approximately 1.5 times wider. This is a property of any sequential test in general. It has to be this way - if the power of a sequential test at a single n were the same as a fixed-n test, then the sequential test would reject just as often, and would therefore produce the same number of false positives under continuous monitoring. The advantage of the sequential test is as follows. The sequential test has many more opportunities to reject the null, not just a single time, and *this leads to stopping experiments sooner on average*.  The following paper has a nice discussion about power debate between fixed-n and sequential tests in section "Power with Optional Stopping — a Tragedy of the Commons?"
> >
> > [2] Grünwald, P., de Heide, R., & Koolen, W. (2019). Safe Testing. arXiv. https://doi.org/10.48550/ARXIV.1906.07801
> >
> > the relevant conclusion here is *optional stopping one needs on average less data to achieve a certain desired power, but one needs to prepare for more data in the worst-case.*.
> >
> >
> >
> > **Lastly, a sequential test usually wouldn't enjoy too much benefit when the considered alternative hypothesis only has a minimum departure from the null**. As described earlier, the average stopping time of the sequential test is less than the corresponding fixed-n test. *But this comparison is already very generous to the fixed-n test* - this evaluation assumes that the unknown parameter is known in the fixed-n test. When doing a fixed-n test one must do a sample size calculation, in which one must specify a minimum detectable effect. Unfortunately, the effect size is unknown. If the statistician were to guess it perfectly, equating the MDE to the true unknown, then a sample size would be determined by the sample size calculation and would be larger, on average, than the stopping time of the sequential test. In practice, however, practitioners set a conservative MDE, which grows their required sample size quadratically. This means that if the practitioner underestimates the true effect size, by specifying a smaller MDE, they would be waiting much longer than they need to, and the fixed-n test would take much longer than the sequential test.
> >
> > Consider our specific example of canary testing. The average stopping time of a sequential test decreases as the effect size increases. Assume we want to both
> > - be able to detect small performance regressions (small effect sizes)
> > - detect large performance regressions (large effect sizes) rapidly.
> >
> > A classical fixed-n test cannot do both, as it can only be applied once. To do the former, a small MDE would be specified, resulting in a very large required sample size. If a huge performance regression occurs, it would go undetected until that pre-specified fixed-n is reached. A sequential test, which doesn't require a sample size calculation, can do both and would catch the performance regression very early.
> >
> > We feel that the concerns and critiques raised are against the approach of sequential testing in general, less about our contributions of a specific sequential test for the considered applications. Our contribution is not to reinvent sequential testing, as this is already a well-established field, with literature dating back to 1940. Our paper is targeting those users who have already decided for themselves that they want to use a sequential procedure, based on the advantages that optional stopping brings, and sets out to contribute a solution.

---

> > > ### Author Response · Authors · 2022-08-09
> > > **Rejecting the alternative fast**
> > >
> > > The further $\theta$ is from $\theta_0$ (KL divergence), the faster the null is rejected. Theorem 2.3 shows that the posterior odds grows asymptotically at a $\exp(n (D_{KL}(\theta | \theta_0))$ rate. The further $\theta$ is from $\theta_0$ in KL divergence, the larger $D_{KL}(\theta | \theta_0)$ is, and the faster the odds process $O_n(\theta_0)$ grows and gains evidence against the null. This means that it crosses the $1/\alpha$ threshold sooner and rejects the null sooner.
> > >
> > > Thank you for your comments about the ambiguity of the notation $O_n(\theta_0)$. We are considering replacing this notation with $O_n^{\theta_0}(x^n)$ to highlight that it is a statistic (a function of the data $x^n$) that is indexed by the specific null hypothesis being tested $\theta_0$. Do you think that would be clearer for the reader?

---

> > > > ### Comment · Reviewer_6qpV · 2022-08-09
> > > > **Thank you for your reply.**
> > > >
> > > > I think my main point is that the main content of this work is spent on an alternative hypothesis with only a small difference from the null. This is a difficult  case, and I didn’t see the incentive that why the proposed test can work well on this difficult case than other sequential tests. Moreover, I don’t consider $\chi^2$ test  as a fair baseline, as it is a fixed-n test and it does not control Type I in the considered setting. It is easy and also it is more fair to frame your problem as a non-parametric sequential two-sample test (Lhéritier, A., & Cazals‘s work mentioned by reviewer pfqx) and use it as a baseline.

---

> > > > > ### Author Response · Authors · 2022-08-09
> > > > > **seeking comparative methods**
> > > > >
> > > > > In our response to reviewer pfqx we explain why the proposed work of Lhéritier, A., & Cazals cannot be directly applied to our problem and reviewer pfqx appears to agree with us. Therefore it is not possible to make this the baseline in a comparison. Is there another sequential test of time inhomogeneous Poisson processes that you can point us to?
> > > > >
> > > > > We’ll restate here for completeness:
> > > > > We don't have any features with which to build a nonparametric regression to predict the multinomial labels. The best we can do is put a dirichlet prior over the unknown label probabilities.
> > > > > Lhéritier, A., & Cazals‘s, their constructions for sequential two-sample tests have some very insightful ideas, but we wouldn't be able to immediately apply their methods in our applications. As we understand it, these papers deal with observations $(l_i, z_i)$ with $z_i \in \mathbb{R}^d$ a feature vector and labels $l_i$ to denote treatment group, and seek to test for differences in distributions of $z_i$'s between groups. In our applications of inhomogeneous Bernoulli and Poisson counting processes we don't have any feature vector to exploit. In the Bernoulli example, all observations for all groups are equal to 1, because we only observe the Bernoulli successes and do not observe the Bernoulli failures. In the Poisson example, all of our observations for all groups are timestamps. In both cases, we have to form a test based on the information available to us, which is simply the number of observations in each treatment group, as the individual observations do not convey any additional information.

---

> > > > > > ### Comment · Reviewer_6qpV · 2022-08-09
> > > > > > **Thank you. My concerns are largely relieved, but categorical variables can be used as features.**
> > > > > >
> > > > > > I think categorical variables can be taken as features. Then a sequential two-sample test is used to test whether group A and group B have homogeneous generation in each categorical variable. But I agree with the authors that this might not be an immediate extension. Moreover, given this is the first sequential test designed to test the multinomial count disparity, I am willing to increase my score to 6.

---

> > > > > > > ### Comment · Reviewer_PFqX · 2022-08-09
> > > > > > > **A bit of confusion**
> > > > > > >
> > > > > > > I am a bit confused about this discussion.
> > > > > > > On one hand, it seems that reviewer 6qpV is referring to Figure 4 (which, by the way, should be mentioned and explained in Section A9 that is empty).
> > > > > > > If I understand correctly, this simulation compares Type I error control between the rejection rule of the proposed sequential multinomial test and the chi-squared test, i.e. in the simple multinomial setup, to illustrate how the sequential procedure keeps type I error under control.
> > > > > > > This case essentially amounts to using a fixed unique cell in the partition-based method of Lhéritier&Cazals 2019 (i.e. there are no features and therefore no feature space to partition).
> > > > > > > On the other hand, the authors seem to refer to the idea of using the sequential nonparametric two-sample test of Lhéritier&Cazals to apply it to the inhomogeneous Bernoulli and Poisson counting processes problems, which I agree that it is not straightforward.

---

> > > > > > > > ### Comment · Reviewer_6qpV · 2022-08-09
> > > > > > > > **The question circles around a fair baseline**
> > > > > > > >
> > > > > > > > I think this question is mainly about seeking a fair baseline. The author chooses $\chi^2$ test as a baseline and it turns out $\chi^2$ does not control Type I error. I don't think controlling Type I error should be claimed as an advantage of a sequential test, because a controlled Type I error should be a basic and first property considered for every two-sample test. Moreover, I still believe the work by Lhéritier can be fairly and easily extended to the multinomial count case -- features become categorical variables and labels become whether the observed categorical variables belong to group A and group B, where group A generates unfair counts and group B generates fair counts. That said, we need to artificially create a group B to generate fair counts and then let the sequential two-sample test to test the difference between group A and group B. This might not be an immediate application but is indeed doable.

---

### Official Review · Reviewer_QXQV · 2022-07-15

**Rating:** 4
**Confidence:** 3
**Soundness:** 3 good
**Presentation:** 2 fair
**Contribution:** 3 good

**Summary:**

This paper introduces two online sequential tests of equality and contrasts, for Bernoulli and Poisson process, based on multinomial hypothesis. In simulated studies, the new tests show the speed up in decision making and reduce the opportunity cost in data collection process.

**Questions:**

- \cite is all wrong in the draft.
- mSPRT-type is not defined
- Figure legends are hard to see clearly


**Limitations:**

adequately, in fact the discussion is thorough.

**Strengths And Weaknesses:**

Strength:
- sequential testing is an important problem and authors demonstrated an effect approach
- some theoretical results are provided on the tests
- multiple applications are tested

Weakness:
- the results are not surprising. For example, theorem 4.1  and Lemma A.3 seems known/straightforward. The resulting test, which is just a ratio test, is not so difficult to come up with.
- no comparison with baselines are conducted in the experiments. It didn't demonstrate the efficiency gain of the proposed approach against potential baselines.

---

> ### Author Response · Authors · 2022-08-01
> **We have comparisons with a baseline test in the supplemental material and our test is distinctly different from likelihood ratio and other ratio tests in terms of its ability to control Type-I error under continuous monitoring**
>
> We would like to thank the reviewer for reading and reviewing our work. We’re grateful for the opportunity to address some of the mentioned weaknesses. We feel that this review does not credit what we feel are our most novel contributions. For example, **weakness (1)** claims *“The resulting test, which is just a ratio test, is not so difficult to come up with”*. Our test is actually distinctly different from a likelihood ratio test, with very different properties. Instead of maximizing the parameters over the alternative parameter space, we obtain a different statistic by instead integrating the parameters in the alternative with respect to a chosen measure. This statistic is a non-negative supermartingale under the null hypothesis, whereas the likelihood ratio test statistic is not. This gives our test a time-uniform Type-I error guarantee, which is not possessed by the likelihood ratio test. That is, our test can be used continuously making optional stopping possible, which is not possible with the likelihood ratio test.
>
> Moreover, coming up with additional test statistics that can be used to test equality in *time inhomogeneous* Bernoulli and Poisson point processes is not our only contribution, the challenge here is how to do this sequentially with time uniform Type I error guarantees. That is, coming up with a test statistic that cancels out time-varying components is only part of the solution, the full solution concerns how to do sequential inference. Existing ratio tests that are alluded to in the feedback do not preserve Type-I error when used repeatedly on an accumulating set of data and we provide a lengthy discussion on why existing approaches are unsatisfactory for our applications in appendix section A.1 “Limitations of Fixed-n Testing”, and motivate our proposal in section A.2 “Solutions via Sequential Testing” (contained in the supplemental material).
>
> Testing for significant differences in *time inhomogeneous* Bernoulli and Poisson counting processes is a very important application in industry. We gave examples of conversion rate optimization, and canary experiments to monitor the release of software, but additional applications occur in many systems that perform real-time data monitoring. These typically operate on real-time streams of data, and it is desired to estimate differences and test for significance in real-time, after every datapoint, without sacrificing statistical guarantees. We believe our work is the first proposal to possess the necessary *anytime-valid* statistical guarantees for these applications, enabling optional stopping within running experiments and providing large savings over existing fixed-n approaches. Due to the ubiquity of these processes in the tech industry and the advantages of our test over existing approaches, we feel that both the scope and the impact of this paper have been potentially underestimated in this review.
>
> To respond to **weakness (2)** about no comparisons - we have in fact included comparisons in the appendix section A.9 Figure 4. The baseline comparison for our test would be a Chi-squared test. Perhaps this is what the reviewer is alluding to as a “ratio test”. We demonstrate that the Chi-squared test does not control Type-I error under continuous monitoring, leading to many false positives if significance is tested frequently, whereas our procedure does control Type-I error, which is expected from our theoretical results. This is what separates our novel contributions from existing approaches. To our knowledge, there are no other tests of time inhomogeneous Bernoulli and Poisson counting processes that provide the same guarantees as ours with which we could provide a further comparison.

---

### Meta-Review · Area_Chair_hJ9K · 2022-08-24

**Recommendation:** Accept
**Confidence:** Less certain

**Metareview:**

While there is no unanimity, the majority is positive and sees the potential value of the approach for machine learning. It is good to mention that the authors did a good job in replying to comments and criticisms, which has clarified a few misunderstandings. Personally I would recommend the authors not to use the confusing terminology between Bayesian and frequentist approaches that are employed in the paper, because it has generated more harm than benefit (at least among those in the discussion here). The theoretical results seem sound and useful and the contribution is good (even if the targeted problem might be seen as too specific by some).

**Award:**

No

---

### Decision · Program_Chairs · 2022-09-14

Accept